# FATE: Feature-Wise Graph Attention with Multi-Period Temporal Encoding for Stock Return Forecasting

## Abstract

Forecasting stock returns remains challenging due to the dual complexity of *temporal heterogeneity* and *structural instability*. On the one hand, stock return series contain signals at multiple horizons: short-term fluctuations, medium-term trends, and long-term cycles. On the other hand, relations across stocks are non-stationary and noisy: correlations strengthen or vanish under sector rotations, liquidity shocks, or macro events. However, existing temporal models often process each stock independently, neglecting cross-stock dependencies and blur signals across horizons. Moreover, structure-aware models typically rely on static or single-view relation-graphs that are brittle under market shifts. To overcome these limitations, we propose *FATE*, which couples *multi-view dynamic graphs* with a *multi-period temporal encoder* to jointly capture cross-stock relations and horizon-specific signals, and further employs *feature-wise graph attention* with a learned graph to integrate these signals coherently while suppressing noise and enhancing interpretability. Experimental results demonstrate that FATE consistently outperforms strong temporal and graph baselines in correlation metrics and investment simulations, which provides more informative and robust signals for stock return forecasting.

## 1 Introduction

Forecasting stock returns has long been a central task in quantitative finance and machine learning (Adam et al., 2016). The challenge stems from the interplay between *heterogeneous temporal patterns* and *non-stationary structural dependencies* in financial markets (West, 1986; Lo & MacKinlay, 1988). Stock return series simultaneously embed signals at multiple horizons: short-term fluctuations often driven by sentiment and trading noise, medium-term trends influenced by sector rotations and capital flows, and long-term cycles linked to macroeconomic fundamentals (Fama & French, 1992; Subrahmanyam, 2010; Harvey et al., 2016). Meanwhile, relations across stocks are highly non-stationary: correlations may strengthen within industries during booms, weaken or vanish under liquidity shocks, and even reverse in response to macro events such as policy changes or interest rate adjustments (Longin & Solnik, 2001; Ang & Chen, 2002; Bernanke & Kuttner, 2005; Chordia et al., 2000; Kyle & Xiong, 2001; Wahal & Yavuz, 2013). For instance, technology stocks may surge together in response to a wave of innovation news, only to decouple when monetary tightening reduces liquidity in growth sectors (Arnell et al., 2023).

While recent studies on the stock return forecasting have shown the potential of capturing sequential patterns and cross-stock dependencies (Kim et al., 2019; Cheng & Li, 2021; Xu et al., 2021; Xiang et al., 2022; Li et al., 2024; Fan & Shen, 2024; Zhu et al., 2024), they are still enduring limitations from relying on single-period encoders and static or single-view graphs, which restrains them from fully exploiting temporal heterogeneity and adapting to evolving market structures.

In terms of temporal modeling, most existing approaches rely on single-period encoders such as GRU, LSTMs, or Transformers (Chung et al., 2014; Hochreiter & Schmidhuber, 1997; Vaswani et al., 2017; Gao, 2016; Rather et al., 2015; Bao et al., 2017). Although effective in capturing sequential dependencies, these models implicitly mix signals across different frequencies, making it difficult to separate transient noise from informative long-term patterns. Moreover, their single-

period design lacks the flexibility to adjust horizon importance under different market regimes, which limits robustness and generalization in volatile financial environments.

In terms of relation-aware modeling, existing methods typically construct graphs from either pre-defined knowledge (e.g., industry sectors or supply chains) or data-driven correlations (Kim et al., 2019; Matsunaga et al., 2019; Feng et al., 2019; Cheng & Li, 2021; Li et al., 2021; Xu et al., 2021; Xiang et al., 2022; Cheng & Li, 2021; Zhao et al., 2023). While these graphs provide useful induc-tive bias, they are usually static or derived from a single view, which prevents them from adapting to rapid changes in market conditions. Moreover, correlation-based graphs often introduce spurious or noisy edges, and integrating multiple heterogeneous relations in a coherent framework remains a challenging problem (Pukthuanthong & Roll, 2015). As a result, relation-aware models still struggle to provide stable and reliable structural information for robust stock return prediction.

These limitations highlight the need for a unified framework that can disentangle multi-horizon tem-poral signals while simultaneously modeling evolving cross-stock relations. To this end, we propose **FATE** (**F**eature-wise graph **A**ttention with multi-period **T**emporal **E**ncoding), which integrates four key components: (i) *multi-view dynamic graph construction*, where evolving relation graphs are derived from price and liquidity features to capture diverse structural patterns; (ii) *multi-period tem-poral encoding*, which disentangles short-, medium-, and long-horizon signals to address temporal heterogeneity; (iii) *graph fusion*, which adaptively integrates multi-view graphs with pairwise corre-lations and market states to form a denoised, robust structure; and (iv) *SuperAttention*, a feature-wise graph attention mechanism applied on the fused sparse graph, enabling coherent integration of tem-poral and structural signals while suppressing noise and enhancing interpretability.

Our main contributions are threefold:

- **Problem analysis.** We identify the dual challenges of temporal heterogeneity and structural insta-bility in stock return forecasting and analyze why existing temporal and relation-aware approaches remain insufficient.
- **Methodology.** We propose **FATE**, a unified framework that couples multi-view evolving rela-tion graphs with a multi-period temporal encoder, and employs feature-wise graph attention on a denoised learned graph to integrate signals coherently while suppressing noise.
- **Evaluation.** On CSI 500 and CSI 1000, FATE consistently outperforms strong baselines across correlation-based metrics and investment simulations, providing richer and more robust signals.

## 2 RELATED WORK

**Temporal modeling.** A large body of work applies sequence models to stock return forecasting, including RNNs, LSTMs, GRUs, and Transformers (Chung et al., 2014; Hochreiter & Schmidhuber, 1997; Vaswani et al., 2017). Some hierarchical and multi-resolution models have been proposed (Franceschi et al., 2019; Duan et al., 2022), but these still lack explicit disentanglement of horizon-specific signals.

**Graph-based modeling.** Models that exploit cross-stock dependencies have become popular. THGNN (Xiang et al., 2022) constructs daily heterogeneous relation graphs from historical prices and uses graph attention to blend relational information. DGDNN (You et al., 2024) proposes dynamic edge construction and graph diffusion to capture time-varying inter-stock dependencies. GRU-PFG (Zhuang et al., 2024) extracts inter-stock correlation from factors using GNNs, perform-ing well even without heavy domain knowledge. FactorGCL (Duan et al., 2025) uses a hypergraph of factors + contrastive learning to capture nonlinear higher-order relations among stocks. While these graph-based methods push performance forward, most rely on static or single-view graphs, risk incorporating noisy or spurious edges, and often lack mechanisms to adaptively fuse or weigh different relational views over time.

**Joint temporal–structural modeling.** Recent works combine temporal and graph-based modules. MASTER (Li et al., 2024) employs a Transformer encoder with cross-time attention to integrate temporal and cross-sectional signals. SuperGATconv (Kim & Oh, 2021) introduces an auxiliary attention loss to improve edge discrimination. Other hybrids such as StockMixer (Fan & Shen, 2024), DGDNN (You et al., 2024), ADGAT (Cheng & Li, 2021), TSRM (Zhao et al., 2023), LSR

(Zhu et al., 2024), HIST (Xu et al., 2021) and FactorGCL (Duan et al., 2025) further illustrate the promise of integrating temporal and structural cues, but they do not explicitly address horizon disentanglement or graph denoising.

# 3 PRELIMINARIES

## 3.1 PROBLEM SETUP

Let $S = \{s_1, s_2, \ldots, s_N\}$ denote a set of $N$ stocks. Each stock $s_i$ is associated with a multivariate historical time series $X_i = \{x_i^1, x_i^2, \ldots, x_i^T\}, x_i^t \in \mathbb{R}^d$, where $x_i^t$ is a $d$-dimensional feature vector observed at trading day $t$. These features typically include price-related variables such as open, high, low, and close (OHLC) prices, as well as *VWAP* (volume weighted average price), *trading volume* (the number of shares traded within a day) and *turnover* (the total monetary value of shares traded, i.e., volume multiplied by price).

The next-day return is defined as $y_i^{t+1} = \frac{p_i^{t+1} - p_i^t}{p_i^t}$, where $p_i^t$ denotes the VWAP of the stock $s_i$ on day $t$. The learning task is to estimate the next-day return given past observations $\{X_i\}$.

## 3.2 DYNAMIC GRAPHS

Stock features are inherently dependent; their co-movements, volatility patterns, and spillover effects, are often correlated due to shared industry sectors, supply-chain linkages, or macroeconomic conditions. Inspired by Xiang et al. (2022), we construct a set of relation graphs that provide complementary views of inter-stock interactions to explicitly capture these dependencies.

Formally, let $\mathcal{G} = \{g_1, g_2, \ldots, g_K\}$ denote $K$ relation graphs. Each graph $g_k \in \mathbb{R}^{N \times N}$ is defined by a correlation-based similarity matrix derived from past trading window. In this work, we construct $K = 6$ relation graphs, each grounded in a distinct market feature with economic interpretation:

- **Opening price graph:** captures the immediate market response to overnight information and pre-market sentiment. Correlations here may reflect the impact of global macro events, policy announcements, or sector-specific news absorbed at the start of the trading day.

- **Closing price graph:** reflects end-of-day consensus valuations, where institutional trading and fundamental factors typically dominate. This graph emphasizes persistent linkages among firms with shared fundamentals or common institutional investor bases.

- **High price graph:** highlights intraday speculative behaviors and extreme upward swings. Strong co-movement at the daily highs may indicate synchronized speculative trading, theme-driven rallies, or short-term momentum spillovers.

- **Low price graph:** characterizes downside co-movements and market stress propagation. Correlations among daily lows reveal how negative shocks, liquidity shortages, or risk aversion transmit across related stocks.

- **Trading volume graph:** captures liquidity linkages through traded quantities. Strong volume correlations indicate simultaneous capital reallocations or liquidity shocks affecting groups of stocks.

- **Turnover graph:** encodes the synchronization of trading activity. Elevated turnover co-movements often signal coordinated portfolio rebalancing, sector rotations, or event-driven strategies that induce simultaneous trading pressure across multiple assets.

By incorporating these six graphs, FATE models different dimensions of stock dependency. Unlike static relation graphs based on industry taxonomies or supply-chain disclosures, our construction is fully data-driven and adapts dynamically to observed market conditions, avoiding the sparsity and noise inherent in manually curated graphs. Together, these graphs provide a richer multi-view representation of evolving market interactions.

Figure 1: FATE includes (i) a multi-period encoder that disentangles horizon-specific signals, (ii) a graph fusion module that integrates dynamic relations, correlations, and market states into $G_{\text{fused}}$, and (iii) a feature-wise graph attention that aggregates signals while suppressing noise.

## 4 FATE FRAMEWORK

### 4.1 OVERVIEW

Figure 1 illustrates the overall architecture of FATE. The framework consists of three main modules: (1) *Dynamic Graph Construction*, (2) *Multi-period Temporal Encoding*, and (3) *Graph-Temporal Fusion and Prediction*. Given historical feature sequences $X$ and multi-view dynamic graphs $\mathcal{G}$, FATE outputs next-day return predictions $\hat{y}_i^{t+1}$ for each stock $s_i \in S$. We outline the core formulation below, with full derivations provided in Appendix A.

### 4.2 MULTI-VIEW DYNAMIC GRAPH CONSTRUCTION

FATE introduces a *multi-view dynamic graph* formulation that extends beyond the static or single-view designs used in prior works. FATE simultaneously constructs six complementary graphs derived from price and liquidity features. This multi-view representation enables the model to disentangle different channels of market interaction, ranging from price co-movements to synchronous trading activities and capital flows.

Formally, for each trading day $t$ we define

$$\mathcal{G}^t = \{g_1^t, g_2^t, \ldots, g_6^t\}, \quad g_k^t[i,j] = \text{Corr}\big(X_i^{t-w:t}[k], X_j^{t-w:t}[k]\big),$$

where $w$ is the rolling window size, $X_i^{t-w:t}[k]$ denotes the feature-$k$ sequence of stock $s_i$, and $g_k^t$ is the adjacency matrix of graph $k$ at time $t$. The dynamic nature of $\mathcal{G}^t$ allows the graph structure to evolve daily, adapting to regime shifts, sector rotations, and liquidity shocks.

The six graph views (open, close, high, low, trading volume, and turnover) were motivated by well-established financial mechanisms and were formalized in Sec. 3.2. Here we emphasize that combining both *price-based* and *liquidity-based* dependencies within a unified graph learning framework is critical: it provides richer structural information than single-view models and offers stronger interpretability by aligning graph semantics with economic intuition.

### 4.3 MULTI-PERIOD TEMPORAL ENCODING

The second key contribution of FATE is a *multi-period temporal encoding* mechanism that explicitly models heterogeneous dependencies across horizons. Unlike a single temporal encoder (e.g., GRU (Chung et al., 2014)) that implicitly mixes frequencies, FATE disentangles *short-term fluctuations*, *medium-term trends*, and *long-term cycles*, aligning with the behavior of short-term traders, institutional investors, and long-horizon funds.

**Temporal partition.** For each stock $s_i$ observed up to time $t$, we partition its $T$-step history into three temporal branches that share the same endpoint $t$ but differ in horizon length:

$$b \in \{\text{short, mid, long}\}, \qquad T_b \in \left\{\frac{T}{4}, \frac{T}{2}, T\right\}.$$

The corresponding segment tensor is $X_b = S[:, -T_b:, :] \in \mathbb{R}^{N \times T_b \times D}$, where $N$ is the number of stocks, $T_b$ the horizon length, and $D$ the raw feature dimension. This design allows the model to capture temporal dynamics at multiple resolutions: the short branch focuses on near-term shocks and local volatility, the mid branch captures sector-level relations and medium-horizon dependencies, and the long branch preserves long-cycle trends driven by macroeconomic conditions.

**Branch encoding.** Following the Transformer architecture (Vaswani et al., 2017), each temporal branch is first linearly embedded into a shared latent space and enriched with sinusoidal positional encodings, which provide temporal order information to otherwise permutation-invariant attention layers. We then apply a standard multi-head self-attention (MHSA) layer followed by a two-layer feed-forward network (FFN) to capture intra-horizon dependencies:

$$H_b = \text{FFN}(\text{MHSA}(\hat{X}_b)), \quad b \in \{\text{short}, \text{mid}, \text{long}\}.$$

This design enables horizon-specific feature extraction rather than mixing signals across frequencies as in single-branch encoders.

**Cross-time aggregation with last-step query.** To reduce each temporal branch to a stock-level summary, we adopt a "last-step query" mechanism (Li et al., 2024), which uses the most recent state to attend over the entire sequence: $H_b = \text{Softmax}\left(\frac{\hat{X}_b[:, -1, :]\hat{X}_b^\top}{\sqrt{F}}\right)\hat{X}_b \in \mathbb{R}^{N \times 1 \times F}$. Squeezing $H_b$ yields three horizon-specific summaries $H_{\text{short}}, H_{\text{mid}}, H_{\text{long}} \in \mathbb{R}^{N \times F}$.

**Adaptive multi-period fusion.** To integrate horizon-specific representations, we first concatenate them into a joint vector: $H_{\text{concat}} = [H_{\text{short}} \| H_{\text{mid}} \| H_{\text{long}}] \in \mathbb{R}^{N \times 3F}$. Then a lightweight two-layer gating network computes raw importance scores, $I = \tanh(H_{\text{concat}}W_1 + \mathbf{b}_1)W_2 + \mathbf{b}_2 \in \mathbb{R}^{N \times 3}$, which are normalized by a softmax across horizons to obtain weights $W \in \mathbb{R}^{N \times 3}$. The final fused representation is given by $H_{\text{fused}} = \sum_{k \in \{\text{short}, \text{mid}, \text{long}\}} W_{:,k} \odot H_k \in \mathbb{R}^{N \times F}$, allowing the model to adaptively emphasize short-term volatility, medium-term sector dynamics, or long-term macroeconomic cycles depending on the market context.

## 4.4 GRAPH FUSION

The third key component of FATE is the *graph fusion module*, which dynamically integrates multiple relational views rather than assigning static weights. The fusion conditions on three complementary factors: (i) dynamic graphs $\mathcal{G}$ derived from market features, (ii) pairwise stock correlations $T$ computed from temporal embeddings, and (iii) market state signals $\beta$ that summarize global conditions.

**Stock correlations.** Given temporal embeddings $H_{\text{fused}} \in \mathbb{R}^{N \times F}$, we obtain stock embeddings $H_{\text{emb}}$ via a two-layer MLP and compute a correlation matrix in an attention-like form:

$$Q = H_{\text{emb}}W_Q, \quad K = H_{\text{emb}}W_K, \quad T = \text{sigmoid}(QK^\top) \in \mathbb{R}^{N \times N},$$

which strengthens edges between stocks with similar temporal patterns. Note that this formulation is inherently asymmetric: $T_{ij} \neq T_{ji}$. Intuitively, $T_{ij}$ reflects how strongly stock $i$ attends to stock $j$, which corresponds to a directed query from $i$ to $j$. In financial graphs, $j$ may be the most informative neighbor of $i$, while the reverse does not necessarily hold. This asymmetry captures the reality that influence between stocks is not always reciprocal, e.g., smaller stocks often attend to the behavior of a large market leader, but the leader does not necessarily react to each individual smaller stock.

**Marke state.** To capture market-wide conditions, we introduce a learnable global market vector $M \in \mathbb{R}^{K \times F}$, which attends to stock embeddings via standard multi-head attention, yielding a contextualized representation that is further projected to graph-specific state weights:

$$M_{\text{env}} = \text{Attention}(M, H_{\text{emb}}, H_{\text{emb}}) \in \mathbb{R}^{K \times F}, \quad \beta = M_{\text{env}}W_\beta \in \mathbb{R}^{K \times 1}.$$

**Graph reweighting and fusion.** We combine dynamic graphs $\mathcal{G}$, pairwise correlations $T$, and market states $\beta$ by reweighting each view with $\beta_k$ and normalizing across $K$ views, then aggregating into a fused structure. To further reduce noisy weak links, we discretize $G_{\text{fused}}$ by a threshold $\tau$.

$$G_{\text{fused}} = \sum_{k=1}^{K} \hat{T}_k \odot g_k, \quad \hat{T}_k = \text{Softmax}_k(\beta_k \cdot T), \quad (G_{\text{binary}})_{ij} = \mathbb{I}(G_{\text{fused}}[i, j] \geq \tau),$$

where $\odot$ denotes element-wise multiplication, $\mathbb{I}(\cdot)$ the indicator function. In financial markets, weak correlations between stocks are spurious and better treated as noise. Retaining only the strong edges provides a cleaner and more reliable structure for downstream message passing.

By jointly considering structural priors ($g_k$), data-driven correlations ($T$), and market-level context ($\beta$), FATE adaptively learns a fused graph that emphasizes informative dependencies and filters out noise, providing a robust structural foundation for return forecasting under different regimes.

### 4.5 SUPERATTENTION

The final module of FATE is *SuperAttention*, a feature-wise graph attention mechanism. Given the fused binary graph and temporal representations, we project them into query, key, and value spaces, $d_h = F/h$ denotes the dimension per head:

$$Q = H_{\text{fused}}W_Q, \quad K = H_{\text{fused}}W_K, \quad V = H_{\text{fused}}W_V.$$

Formally, the attention weights for the stock $i$ are computed around its neighbor as

$$A_{ij} = \text{Softmax}_{j \in \mathcal{N}(i)}\left(\frac{Q_i K_j^\top}{\sqrt{d_h}}\right), \quad \mathcal{N}(i) = \{j \mid G_{\text{binary}}[i,j] = 1\}.$$

**Gating mechanism.** To adaptively modulate feature contributions, we introduce a feature-wise gating function (Cheng & Li, 2021) that depends on both the target stock $i$ and the neighbor $j$. For each stock pair $(i,j)$, we first construct a relation feature by concatenating their value vectors: $C_{ij} = [V_i \| V_j] \in \mathbb{R}^{2d_h}$. The gating vector is then generated as $g_{ij} = \tanh(C_{ij}W_g + \mathbf{b}_g) \in \mathbb{R}^{d_h}$. The neighbor representation $V_j$ is modulated by $g_{ij}$: $\tilde{V}_{ji} = V_j \odot g_{ij} \in \mathbb{R}^{d_h}$.

This mechanism conditions on both $V_i$ and $V_j$ and selectively filters individual feature of $V_j$ before aggregation. In this way, features of $V_j$ that are irrelevant or noisy to $i$ are suppressed, while informative ones are retained, leading to a cleaner and more targeted message-passing process.

**Aggregation.** The output for node $i$ is obtained by attending over its neighbors with sparse weights and feature-wise gates, and multi-head results are concatenated and projected back to $F$:

$$H_{\text{out}} = \text{Concat}_h\left(\sum_{j \in \mathcal{N}(i)} A_{ij}^{(h)} \tilde{V}_{ji}^{(h)}\right)W_O, \quad W_O \in \mathbb{R}^{F \times F}.$$

Compared with standard GAT (Velickovic et al., 2017), which assigns a single scalar weight to each neighbor, SuperAttention incorporates: (i) *sparse edge masking*, enforcing attention only along valid edges in $G_{\text{binary}}$; (ii) *feature-wise gating*, which modulates the importance of each feature in a neighbor's embedding. This design reduces noise, improves interpretability, and yields more robust graph representations under varying market regimes.

### 4.6 RETURN FORECASTING AND GRAPH-REGULARIZED LOSS

With both the temporal encoding $H_{\text{fused}}$ and the graph encoding $H_{\text{out}}$, we concatenate them and feed into a simple MLP to predict the next-day return: $\hat{y} = \text{MLP}([H_{\text{fused}} \| H_{\text{out}}])$.

**Regression loss.** The primary objective is to minimize the prediction error between the ground truth $y_i$ and the estimated return $\hat{y}_i$. We adopt mean squared error: $\mathcal{L}_{\text{MSE}} = \frac{1}{N}\sum_{i=1}^{N}(y_i - \hat{y}_i)^2$.

**Graph-regularized loss.** Inspired by SuperGAT (Kim & Oh, 2021), we introduce edge-level supervision to regularize attention weights. For each edge $(i,j)$ in the graph, we compute the edge-level attention-derived probability that $i$ attends to $j$: $p_{ij} = \text{sigmoid}(Q_i K_j^\top)$. This asymmetry captures non-reciprocal influence (e.g., smaller-cap stocks attend to large-cap leaders, but not vice versa). Observed edges $(i,j) \in E^+$ are treated as positive samples, while non-edges $(i,j) \in E^-$ are negatives. The auxiliary edge loss is defined as binary cross-entropy:

$$\mathcal{L}_E = -\frac{1}{|U|}\sum_{(i,j) \in U}\left[\mathbb{I}_{(i,j) \in E^+} \cdot \log p_{ij} + \mathbb{I}_{(i,j) \in E^-} \cdot \log(1 - p_{ij})\right],$$

where $U \subseteq E^+ \cup E^-$ is a sampled mini-batch of edges.

**Overall objective.** The final training loss is $\mathcal{L} = \mathcal{L}_{\text{MSE}} + \lambda_E \mathcal{L}_E$, where $\lambda_E$ balances predictive accuracy and structural regularization. In practice, $\mathcal{L}_E$ dominates the early phase of training, rapidly guiding the model toward a plausible graph structure. As training proceeds, the optimization gradually shifts its focus to $\mathcal{L}_{\text{MSE}}$, which fine-tunes the model for accurate return forecasting.

**Differentiable binary graph via STE.** Note that applying a binary operation on $G_{\text{fused}}$ will cause gradient vanishing, since the thresholding indicator $\mathbb{I}(\cdot)$ is non-differentiable. To address this, we employ the *straight-through estimator* (STE) (Courbariaux et al., 2016), which decouples the forward and backward passes:

$$G_{\text{binary}} = \mathbb{I}(G_{\text{fused}} \geq \tau) + G_{\text{fused}} - G_{\text{fused}}^{\text{detach}}.$$

Here $G_{\text{fused}}^{\text{detach}}$ denotes a gradient-detached copy. Thus, in the forward pass the model uses a strict binary adjacency, while in the backward pass gradients are directly propagated through $G_{\text{fused}}$.

This mechanism yields a *differentiable binary graph*: it combines the interpretability of a hard adjacency with the trainability of a continuous one, allowing FATE to denoise weak edges while still refining the underlying graph structure through backpropagation.

## 5 EXPERIMENTS

### 5.1 EXPERIMENTAL SETTING

**Datasets and Preprocessing.** We evaluate FATE on constituents of CSI 500 and CSI 1000 from 2017-01-01 to 2024-12-31. For each stock and trading day $t$, we construct a 60-day lookback with daily features: open, high, low, close, VWAP, and trading volume. All features are cross-sectionally standardized per day to remove market-level scale shifts. Labels are computed as next-day returns, i.e., the daily percentage change of VWAP, and then rank-standardized within the day to form a robust supervision target. The dynamic graphs are constructed following the procedure in Sec. 3.2.

**Walk-forward evaluation.** We adopt an expanding walk-forward protocol with strictly consecutive splits: each fold consists of 4 years for training, immediately followed by 1 year for validation, and the next 1 year for testing. The window slides forward by one year, producing three evaluation folds whose test periods are 2022, 2023, and 2024. We report the mean and standard deviation across folds and random seeds.

**Metrics.** We evaluate models with standard correlation-based metrics: **IC** (Pearson) (Pearson, 1896) and **RankIC** (Spearman) (Spearman, 1961), averaged over the test horizon. Stability is measured by **ICIR** and **RankICIR** (mean divided by standard deviation) (Treynor & Black, 1973; Grinold et al., 1989), and portfolio relevance by **ICW**, which upweights top-decile predictions. All results are reported as mean (standard deviation) across folds and seeds.

**Baselines.** We compare FATE with representative methods including temporal models: MLP (LeCun et al., 2015), LSTM (Hochreiter & Schmidhuber, 1997), GRU (Chung et al., 2014), Transformer (Vaswani et al., 2017); graph-based (static graph included) models: HIST (Xu et al., 2021), THGNN (Xiang et al., 2022), HATS (Kim et al., 2019), TSRM (Zhao et al., 2023); and classic models: FactorVAE (Duan et al., 2022), StockMixer (Fan & Shen, 2024), FactorGCL (Duan et al., 2025), MASTER Li et al. (2024).

**Implementation Details** Implementation Details. We implement our framework with PyTorch and conduct all experiments on a system equipped with eight NVIDIA GeForce RTX 4090 GPUs.

The hyper-parameter settings and the introduction of baselines are provided in Appendix B.

### 5.2 MAIN RESULTS

Tables 1 and 2 summarize the performance of all methods on CSI 500 and CSI 1000, respectively. Across both benchmarks, FATE consistently achieves the best results on all correlation-based metrics, outperforming strong baselines. The improvements are particularly pronounced on CSI 1000,

Table 1: Performance comparison on CSI 500. We report mean (std) across three folds (test year 2022-2024) and ten seeds. Best results are in bold.

| Method | IC | RankIC | ICIR | RankICIR | ICW |
|---|---|---|---|---|---|
| MLP | 0.0407 (0.0020) | 0.0383 (0.0023) | 0.2642 (0.0210) | 0.2498 (0.0246) | 0.0516 (0.0047) |
| GRU | 0.0413 (0.0032) | 0.0371 (0.0047) | 0.2797 (0.0140) | 0.2610 (0.0255) | 0.0505 (0.0035) |
| LSTM | 0.0439 (0.0038) | 0.0303 (0.0037) | 0.2776 (0.0242) | 0.2412 (0.0257) | 0.0539 (0.0048) |
| Transformer | 0.0364 (0.0046) | 0.0314 (0.0061) | 0.2459 (0.0275) | 0.2113 (0.0366) | 0.0434 (0.0061) |
| HIST | 0.0399 (0.0049) | 0.0312 (0.0058) | 0.2593 (0.0330) | 0.2063 (0.0338) | 0.0473 (0.0055) |
| FactorVAE | 0.0345 (0.0067) | 0.0352 (0.0056) | 0.2437 (0.0600) | 0.2132 (0.0416) | 0.0425 (0.0071) |
| THGNN | 0.0278 (0.0062) | 0.0245 (0.0063) | 0.1783 (0.0489) | 0.1577 (0.0484) | 0.0286 (0.0076) |
| StockMixer | 0.0243 (0.0057) | 0.0122 (0.0061) | 0.1497 (0.0541) | 0.1321 (0.0515) | 0.0136 (0.0077) |
| HATS | 0.0406 (0.0040) | 0.0369 (0.0044) | 0.2554 (0.0370) | 0.2401 (0.0476) | 0.0469 (0.0056) |
| FactorGCL | 0.0389 (0.0050) | 0.0311 (0.0058) | 0.2296 (0.0340) | 0.2153 (0.0388) | 0.0493 (0.0060) |
| TSRM | 0.0426 (0.0041) | 0.0307 (0.0047) | 0.2550 (0.0410) | 0.2423 (0.0355) | 0.0535 (0.0065) |
| MASTER | 0.0451 (0.0039) | 0.0425 (0.0040) | 0.3143 (0.0395) | 0.2920 (0.0330) | 0.0570 (0.0062) |
| **FATE** | **0.0521** (0.0032) | **0.0491** (0.0033) | **0.3376** (0.0310) | **0.3192** (0.0276) | **0.0619** (0.0057) |

Table 2: Performance comparison on CSI 1000. We report mean (std) across three folds (test year 2022-2024) and ten seeds. Best results are in bold.

| Method | IC | RankIC | ICIR | RankICIR | ICW |
|---|---|---|---|---|---|
| MLP | 0.0502 (0.0037) | 0.0519 (0.0048) | 0.3825 (0.0264) | 0.3384 (0.0273) | 0.0595 (0.0052) |
| GRU | 0.0585 (0.0035) | 0.0569 (0.0035) | 0.4046 (0.0398) | 0.3840 (0.0269) | 0.0688 (0.0039) |
| LSTM | 0.0507 (0.0019) | 0.0495 (0.0016) | 0.3545 (0.0183) | 0.3458 (0.0171) | 0.0600 (0.0024) |
| Transformer | 0.0608 (0.0034) | 0.0586 (0.0033) | 0.4200 (0.0304) | 0.3918 (0.0158) | 0.0699 (0.0035) |
| HIST | 0.0573 (0.0020) | 0.0538 (0.0026) | 0.4119 (0.0162) | 0.3612 (0.0191) | 0.0647 (0.0030) |
| FactorVAE | 0.0553 (0.0020) | 0.0554 (0.0029) | 0.4432 (0.0185) | 0.4031 (0.0104) | 0.0605 (0.0022) |
| THGNN | 0.0527 (0.0023) | 0.0512 (0.0024) | 0.3627 (0.0236) | 0.3495 (0.0217) | 0.0586 (0.0015) |
| StockMixer | 0.0411 (0.0055) | 0.0395 (0.0052) | 0.2970 (0.0312) | 0.2927 (0.0300) | 0.0443 (0.0058) |
| HATS | 0.0587 (0.0033) | 0.0565 (0.0038) | 0.3829 (0.0438) | 0.3620 (0.0339) | 0.0684 (0.0040) |
| FactorGCL | 0.0532 (0.0034) | 0.0506 (0.0034) | 0.4002 (0.0285) | 0.3841 (0.0154) | 0.0611 (0.0032) |
| TSRM | 0.0562 (0.0037) | 0.0554 (0.0040) | 0.3452 (0.0265) | 0.3417 (0.0164) | 0.0642 (0.0035) |
| MASTER | 0.0627 (0.0028) | 0.0618 (0.0030) | 0.4470 (0.0243) | 0.4036 (0.0161) | 0.0739 (0.0030) |
| **FATE** | **0.0668** (0.0013) | **0.0645** (0.0011) | **0.4728** (0.0221) | **0.4254** (0.0165) | **0.0765** (0.0015) |

where market dynamics are noisier and small-cap stocks are more sensitive to liquidity shocks: FATE improves IC by around 0.8% absolute on CSI 500 and 0.6% absolute on CSI 1000, and boosts ICIR and RankICIR by a significant margin. These results demonstrate that explicitly disentangling temporal horizons and adaptively fusing multi-view relations provide more robust and informative signals for stock return prediction, especially in challenging market segments.

## 5.3 ABLATION STUDY

We evaluate the impact of each module by removing or simplifying it. Results in Tables 3 and 4 show that (1) replacing multi-period encoders with a single-period encoder or removing horizon fusion significantly reduces IC/ICIR, (2) simplifying graph fusion (mean graph, no stock correlations, no market states) lowers RankIC/RankICIR, and (3) removing feature-wise gating or graph-regularized loss (GRLoss) weakens ICW and stability. Overall, each component contributes, and full FATE achieves the best performance across both datasets.

## 5.4 PORTFOLIO BACKTESTING

We simulate a daily rebalanced long-only top-$k$ portfolio. At each trading day, we rank all stocks by predicted scores, select the top-$k$, allocate portfolio weights using a linear-decay scheme, and rebalance by liquidating all prior positions and buying the updated top-$k$ stocks. Transaction costs are set to 0.05% per buy and 0.10% per sell, reflecting realistic frictions in the market.

Table 3: Ablation study results on CSI 500. Mean (std) across folds and seeds.

| Variant | IC | RankIC | ICIR | RankICIR | ICW |
|---|---|---|---|---|---|
| FATE (ours) | **0.0521** (0.0032) | **0.0491** (0.0033) | **0.3376** (0.0310) | **0.3192** (0.0276) | **0.0619** (0.0057) |
| SinglePeriod | 0.0416 (0.0060) | 0.0323 (0.0067) | 0.2399 (0.0320) | 0.2297 (0.0400) | 0.0492 (0.0069) |
| NoPeriodFuse | 0.0431 (0.0049) | 0.0367 (0.0044) | 0.2494 (0.0261) | 0.2376 (0.0310) | 0.0500 (0.0058) |
| MeanGraph | 0.0444 (0.0061) | 0.0371 (0.0054) | 0.2978 (0.0330) | 0.2777 (0.0334) | 0.0539 (0.0065) |
| NoCorrelation | 0.0451 (0.0057) | 0.0417 (0.0045) | 0.2857 (0.0287) | 0.2653 (0.0311) | 0.0519 (0.0060) |
| NoMarket | 0.0467 (0.0057) | 0.0435 (0.0049) | 0.2898 (0.0300) | 0.2799 (0.0299) | 0.0547 (0.0060) |
| NoGate | 0.0511 (0.0055) | 0.0476 (0.0049) | 0.3021 (0.0293) | 0.2937 (0.0450) | 0.0606 (0.0066) |
| NoGRLoss | 0.0507 (0.0060) | 0.0462 (0.0054) | 0.3006 (0.0374) | 0.2998 (0.0440) | 0.0596 (0.0068) |

Table 4: Ablation study results on CSI 1000. Mean (std) across folds and seeds.

| Variant | IC | RankIC | ICIR | RankICIR | ICW |
|---|---|---|---|---|---|
| FATE (ours) | **0.0668** (0.0013) | **0.0645** (0.0011) | **0.4728** (0.0221) | **0.4254** (0.0165) | **0.0765** (0.0015) |
| SinglePeriod | 0.0600 (0.0024) | 0.0589 (0.0023) | 0.4118 (0.0291) | 0.3771 (0.0180) | 0.0683 (0.0031) |
| NoPeriodFuse | 0.0610 (0.0021) | 0.0592 (0.0024) | 0.4345 (0.0230) | 0.3978 (0.0163) | 0.0698 (0.0031) |
| MeanGraph | 0.0611 (0.0025) | 0.0594 (0.0024) | 0.4338 (0.0227) | 0.3989 (0.0170) | 0.0703 (0.0034) |
| NoCorrelation | 0.0619 (0.0023) | 0.0603 (0.0025) | 0.4367 (0.0250) | 0.3879 (0.0172) | 0.0702 (0.0033) |
| NoMarket | 0.0628 (0.0026) | 0.0599 (0.0024) | 0.4178 (0.0271) | 0.3888 (0.0157) | 0.0704 (0.0032) |
| NoGate | 0.0631 (0.0016) | 0.0615 (0.0020) | 0.4283 (0.0178) | 0.4031 (0.0159) | 0.0719 (0.0029) |
| NoGRLoss | 0.0630 (0.0023) | 0.0613 (0.0025) | 0.4353 (0.0240) | 0.4052 (0.0150) | 0.0731 (0.0040) |

We use **cumulative return (CR)** as the main evaluation metric, defined as $\text{CR} = \frac{V_T}{V_0}$, where $V_T$ is the final portfolio value and $V_0 = 1e8$ is the initial capital. To determine $k$, we perform grid search on the validation set with $k \in \{20, 30, 40, 50, 100, 120, 150\}$, selecting $k = 50$ for CSI 500 and $k = 100$ for CSI 1000. FATE achieves $\sim$40% cumulative return on CSI 500 and $\sim$80% on CSI 1000 during 2022–2024, significantly outperforming strong baselines. Figure 2 illustrates the cumulative return trajectories.

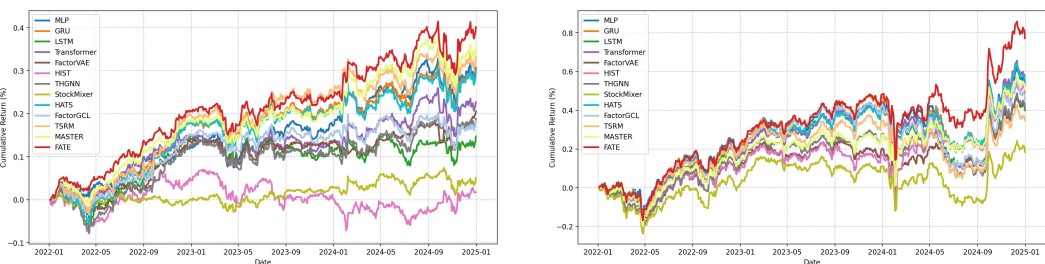

Figure 2: Cumulative Return on CSI 500 (left) CSI 1000 (right) from 2022 to 2024.

## 6 CONCLUSION

We presented **FATE**, a unified framework for stock return forecasting that addresses two long-standing challenges in financial prediction: temporal heterogeneity and structural instability. By disentangling short-, mid-, and long-horizon signals with a multi-period encoder, adaptively integrating multi-view relation graphs through graph fusion, and applying feature-wise SuperAttention with structural regularization, FATE achieves state-of-the-art performance on CSI 500 and CSI 1000. Extensive experiments and ablations confirm that each component contributes to both predictive accuracy and interpretability. Beyond outperforming temporal and graph-based baselines, FATE provides a general template for learning from multi-horizon dynamics and evolving relations in other noisy, non-stationary domains.

**Reproducibility Statement.** We have made every effort to ensure reproducibility by providing detailed model formulations, training protocols, and evaluation settings in the main text and appendix. All code, preprocessing scripts, and processed CSI 500/1000 datasets are included in the supplementary materials, enabling end-to-end reproduction of our results.

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

# A  FATE FRAMEWORK DETAILED DERIVATIONS

## A.1  MULTI-PERIOD TEMPORAL ENCODING

The second key contribution of FATE is a *multi-period temporal encoding* mechanism that explicitly models heterogeneous dependencies across horizons. Unlike a single temporal encoder (e.g., RNN/Transformer) that implicitly mixes frequencies, FATE disentangles *short-term fluctuations*, *medium-term trends*, and *long-term cycles*, aligning with the behavior of short-term traders, institutional investors, and long-horizon funds.

**Temporal partition.**  For each stock $s_i$ observed up to time $t$, we partition its $T$-step history into three temporal branches that share the same endpoint $t$ but differ in horizon length:

$$b \in \{\text{short, mid, long}\}, \qquad T_b \in \left\{ \tfrac{T}{4},\ \tfrac{T}{2},\ T \right\}.$$

The corresponding segment tensor is $X_b = S[:,\ -T_b:,\ :\ ] \in \mathbb{R}^{N \times T_b \times D}$, where $N$ is the number of stocks, $T_b$ the horizon length, and $D$ the raw feature dimension. This design allows the model to capture temporal dynamics at multiple resolutions: the short branch focuses on near-term shocks and local volatility, the mid branch captures sector-level relations and medium-horizon dependencies, and the long branch preserves long-cycle trends driven by macroeconomic conditions.

**Linear embedding and positional encoding.**  Since the raw features in $X_b$ have width $D$, each branch is first projected into a common latent dimension $F$ by a learnable linear embedding:

$$Z_b = X_b W_{\text{in}} + \mathbf{b}_{\text{in}}, \qquad W_{\text{in}} \in \mathbb{R}^{D \times F},\ \ \mathbf{b}_{\text{in}} \in \mathbb{R}^{F},$$

yielding $Z_b \in \mathbb{R}^{N \times T_b \times F}$. This transformation unifies all horizons into the same embedding space and enables the network to learn expressive linear combinations of raw market features.

Since self-attention is permutation-invariant, we add temporal order information via a sinusoidal positional encoding $P_b \in \mathbb{R}^{T_b \times F}$ shared across stocks:

$$(P_b)_{t,2j} = \sin\!\Big( \tfrac{t}{10000^{2j/F}} \Big), \qquad (P_b)_{t,2j+1} = \cos\!\Big( \tfrac{t}{10000^{2j/F}} \Big), \quad t = 0, \ldots, T_b - 1.$$

To apply this encoding consistently across all $N$ stocks, we replicate it along the batch axis:

$$\hat{X}_b = Z_b + \mathbf{1}_N \otimes P_b \ \in\ \mathbb{R}^{N \times T_b \times F},$$

where $\mathbf{1}_N \otimes P_b$ denotes tiling of $P_b$ for each stock. The resulting sequence $\hat{X}_b$ thus integrates both feature semantics and temporal order, and is used as the input to the subsequent multi-head self-attention encoder.

**Multi-head self-attention and FFN.**  To model intra-horizon dependencies, each temporal branch is encoded using multi-head self-attention. Let the number of heads be $h$ and the per-head width $d_h = F/h$. For branch $b$:

$$Q_b^h = \hat{X}_b W_q^h, \quad K_b^h = \hat{X}_b W_k^h, \quad V_b^h = \hat{X}_b W_v^h,$$

where $W_q^h, W_k^h, W_v^h \in \mathbb{R}^{F \times d_h}$ are learnable matrices. Scaled dot-product attention is then

$$A_b^h = \text{Softmax}\!\Big( \tfrac{Q_b^h (K_b^h)^\top}{\sqrt{d_h}} \Big) \in \mathbb{R}^{N \times T_b \times T_b},$$

$$\text{head}_h = A_b^h V_b^h \in \mathbb{R}^{N \times T_b \times d_h}.$$

The head outputs are concatenated and projected back to width $F$:

$$H_b = \text{Concat}(\text{head}_1, \ldots, \text{head}_h)\, W_o, \qquad W_o \in \mathbb{R}^{h d_h \times F}, \quad H_b \in \mathbb{R}^{N \times T_b \times F}.$$

To enrich feature transformations, we apply a position-wise feedforward network to each time step:

$$\text{FFN}(x) = \text{ReLU}(x W_1 + \mathbf{b}_1)\, W_2 + \mathbf{b}_2 + x,$$

with parameters $W_1 \in \mathbb{R}^{F \times F_f}, W_2 \in \mathbb{R}^{F_f \times F}, \mathbf{b}_1 \in \mathbb{R}^{F_f}, \mathbf{b}_2 \in \mathbb{R}^{F}$.

# B    BASELINES AND IMPLEMENTATION DETAILS

We compare our proposed framework against a broad set of stock return forecasting methods:

- **MLP** (LeCun et al., 2015): a three-layer multilayer perceptron.
- **LSTM** (Hochreiter & Schmidhuber, 1997): a recurrent model based on long short-term memory units.
- **GRU** (Chung et al., 2014): a recurrent model using gated recurrent units.
- **Transformer** (Vaswani et al., 2017): a self-attention based sequence model.
- **FactorVAE** (Duan et al., 2022): a VAE-based factor model for financial features.
- **StockMixer** (Fan & Shen, 2024): an MLP-based model that mixes features across temporal, cross-sectional, and stock dimensions.
- **HIST** (Xu et al., 2021): a hypergraph model combining predefined concepts (e.g., industries) with latent concepts.
- **THGNN** (Xiang et al., 2022): a temporal hypergraph neural network that builds stock relation graphs from historical price–volume features.
- **HATS** (Kim et al., 2019): a relation-aware hypergraph model that leverages structured market information to generate relation embeddings.
- **FactorGCL** (Duan et al., 2025): a factor-based model using a residual graph convolutional structure.
- **TSRM** (Zhao et al., 2023): a model constructing stock relation graphs by clustering historical closing-price correlations.
- **MASTER** (Li et al., 2024): a stock Transformer that captures momentary cross-time correlations and uses market data for adaptive feature selection

These baselines cover temporal, factor-based, and graph-based approaches, providing a comprehensive comparison.

We use Adam with learning rates in $\{1e{-}3, 5e{-}4, 2e{-}4, 1e{-}4\}$, early stopping on validation IC, weight decay $1e{-}5$, dropout 0.1, and gradient clipping at 1.0. We repeat all experiments with five random seeds and report mean $\pm$ standard deviation. Mini-batches correspond to all stocks on a day, so batch size equals the cross-sectional stock count. For FATE we set the three temporal windows to $\{15, 30, 60\}$ by default (ablation in Sec. 5.3), graph threshold $\tau \in [0.5, 0.9]$ (validated), and edge-regularization weight $\lambda_E \in \{0.1, 0.2, 0.5, 1.0\}$.

The weighted information coefficient (ICW), which upweights the top decile of predicted scores. Let $w_i$ be the sample weight defined as

$$w_i = \begin{cases} 2, & \hat{y}_i \geq Q_{0.9}(\hat{y}), \\ 1, & \text{otherwise.} \end{cases}$$

The weighted means of $y$ and $\hat{y}$ are

$$\bar{y}_w = \frac{\sum_{i=1}^{n} w_i y_i}{\sum_{i=1}^{n} w_i}, \qquad \bar{\hat{y}}_w = \frac{\sum_{i=1}^{n} w_i \hat{y}_i}{\sum_{i=1}^{n} w_i}.$$

The weighted covariance and variances are

$$\text{Cov}_w(y, \hat{y}) = \frac{\sum_{i=1}^{n} w_i (y_i - \bar{y}_w)(\hat{y}_i - \bar{\hat{y}}_w)}{\sum_{i=1}^{n} w_i},$$

$$\text{Var}_w(y) = \frac{\sum_{i=1}^{n} w_i (y_i - \bar{y}_w)^2}{\sum_{i=1}^{n} w_i}, \qquad \text{Var}_w(\hat{y}) = \frac{\sum_{i=1}^{n} w_i (\hat{y}_i - \bar{\hat{y}}_w)^2}{\sum_{i=1}^{n} w_i}.$$

Finally, the weighted IC is computed as

$$\rho_w = \frac{\text{Cov}_w(y, \hat{y})}{\sqrt{\text{Var}_w(y)\,\text{Var}_w(\hat{y})}}.$$

Table 5: Performance comparison on SPX 500. We report mean (std) across three folds (test years 2022–2024) and ten seeds. Best results are in bold.

| Method | IC | RankIC | ICIR | RankICIR | ICW |
|---|---|---|---|---|---|
| MLP | 0.0161 (0.0025) | 0.0156 (0.0028) | 0.1081 (0.0185) | 0.1197 (0.0201) | 0.0113 (0.0032) |
| GRU | 0.0329 (0.0031) | 0.0285 (0.0035) | 0.1715 (0.0202) | 0.1693 (0.0236) | 0.0288 (0.0041) |
| LSTM | 0.0309 (0.0034) | 0.0225 (0.0038) | 0.1707 (0.0235) | 0.1538 (0.0252) | 0.0205 (0.0043) |
| Transformer | 0.0363 (0.0028) | 0.0319 (0.0032) | 0.1908 (0.0193) | 0.1605 (0.0214) | 0.0399 (0.0038) |
| HIST | 0.0339 (0.0030) | 0.0341 (0.0034) | 0.1975 (0.0212) | 0.1653 (0.0228) | 0.0403 (0.0036) |
| FactorVAE | 0.0193 (0.0036) | 0.0177 (0.0040) | 0.1249 (0.0251) | 0.1282 (0.0267) | 0.0200 (0.0045) |
| THGNN | 0.0340 (0.0029) | 0.0339 (0.0033) | 0.1952 (0.0206) | 0.1843 (0.0222) | 0.0368 (0.0039) |
| StockMixer | 0.0312 (0.0032) | 0.0291 (0.0036) | 0.1600 (0.0227) | 0.1513 (0.0243) | 0.0258 (0.0042) |
| HATS | 0.0319 (0.0031) | 0.0297 (0.0035) | 0.1785 (0.0202) | 0.1753 (0.0236) | 0.0389 (0.0044) |
| FactorGCL | 0.0399 (0.0027) | 0.0374 (0.0033) | 0.2785 (0.0192) | 0.2344 (0.0202) | 0.0479 (0.0049) |
| TSRM | 0.0359 (0.0029) | 0.0344 (0.0033) | 0.1982 (0.0206) | 0.1776 (0.0222) | 0.0407 (0.0039) |
| MASTER | 0.0379 (0.0028) | 0.0334 (0.0032) | 0.1985 (0.0198) | 0.1753 (0.0214) | 0.0429 (0.0038) |
| **FATE** | **0.0439** (0.0023) | **0.0404** (0.0027) | **0.2985** (0.0163) | **0.2753** (0.0179) | **0.0493** (0.0032) |

## C  ADDITIONAL EXPERIMENTS AND ANALYSES

### C.1  CROSS-MARKET GENERALIZATION VERIFICATION

We further assess the cross-market generalization ability of FATE by evaluating it on the SPX 500 universe. This setup tests whether a model trained on CSI-style data can retain performance under different market characteristics. As shown in Table 5, FATE consistently outperforms all baselines, confirming its robust cross-market predictive capability.

### C.2  SUPPLEMENTARY PORTFOLIO-LEVEL EVALUATION

To complement predictive metrics such as IC and IR, we report portfolio-level performance using two standard indicators in quantitative finance: *Sharpe Ratio* and *Maximum Drawdown (MDD)*. Sharpe Ratio measures risk-adjusted return,

$$\text{Sharpe} = \frac{\mathbb{E}[R_t]}{\sqrt{\text{Var}(R_t)}},$$

where $\{R_t\}_{t=1}^{T}$ is the daily portfolio return series.

Let $\{P_t\}_{t=1}^{T}$ denote the cumulative portfolio value over time, computed from the daily returns. Maximum Drawdown (MDD) measures the largest peak-to-trough decline during the evaluation period, and is formally defined as

$$\text{MDD} = \max_{1 \leq t_2 < t_1 \leq T} \frac{P_{t_2} - P_{t_1}}{P_{t_2}},$$

where $P_{t_2}$ is a historical peak and $P_{t_1}$ is a subsequent trough. A larger MDD indicates a more severe downside risk. Both metrics capture complementary aspects of portfolio behavior: Sharpe Ratio evaluates stability of returns, whereas MDD reflects downside risk.

Table 6 summarizes the results. Across both universes, FATE achieves the highest Sharpe Ratio among all baselines, indicating improved risk-adjusted performance. In terms of Maximum Drawdown, FATE remains broadly comparable to strong competitors: although its MDD is not always the lowest, it stays within a similar range as other high-performing models (e.g., MASTER and HIST). This suggests that the return improvements brought by FATE do not come at the cost of substantially increased downside exposure, yielding a balanced and robust portfolio profile overall.

### C.3  ABLATION STUDY ON THE MULTI-VIEW GRAPH CONSTRUCTION

To assess the contribution of each feature view in the multi-view dynamic graph, we conduct ablation experiments by removing one view at a time from the set {open, close, high, low, volume, turnover}.

Table 6: Sharpe Ratio and Maximum Drawdown on CSI 500 and CSI 1000.

| | CSI 500 | | CSI 1000 | |
| Method | Sharpe | MDD | Sharpe | MDD |
|---|---|---|---|---|
| MLP | 1.2796 | 0.1065 | 0.9163 | **0.2698** |
| GRU | 1.2354 | 0.0994 | 0.9381 | 0.3123 |
| LSTM | 0.6913 | 0.0948 | 0.7496 | 0.3272 |
| Transformer | 0.9795 | 0.1123 | 0.9147 | 0.2940 |
| FactorVAE | 1.0529 | 0.0660 | 0.7769 | 0.2924 |
| HIST | 1.1160 | 0.1410 | 0.8483 | 0.3235 |
| THGNN | 0.8175 | 0.1076 | 0.7289 | 0.2701 |
| StockMixer | 0.3605 | **0.0628** | 0.4182 | 0.2797 |
| HATS | 1.1338 | 0.1018 | 0.9320 | 0.2840 |
| FactorGCL | 0.8640 | 0.0842 | 0.7955 | 0.3502 |
| TSRM | 1.2435 | 0.1073 | 0.6695 | 0.3284 |
| MASTER | 1.4410 | 0.1330 | 0.8929 | 0.2906 |
| **FATE** | **1.5606** | 0.1239 | **1.1748** | 0.3030 |

These views describe different aspects of price movement and liquidity, and although they are not fully independent (e.g., volume and turnover can show mild correlation in practice), each view provides useful information to the overall graph structure. As shown in Tables 7 and 8, removing any single view leads to a consistent performance drop on both CSI 500 and CSI 1000, indicating that every view contributes positively to the representation. The best performance is achieved when all views are included, demonstrating that the full multi-view construction offers the richest description of stock relationships and is therefore adopted as our default setting.

Table 7: Ablation study on the multi-view graph construction on CSI 500.

| Method | IC | RankIC | ICIR | RankICIR | ICW |
|---|---|---|---|---|---|
| No_Open | 0.0511 (0.0031) | 0.0458 (0.0033) | 0.3002 (0.0333) | 0.3012 (0.0331) | 0.0610 (0.0060) |
| No_Close | 0.0519 (0.0030) | 0.0462 (0.0031) | 0.3142 (0.0330) | 0.2999 (0.0303) | 0.0600 (0.0059) |
| No_High | 0.0510 (0.0033) | 0.0448 (0.0033) | 0.3022 (0.0311) | 0.3007 (0.0260) | 0.0607 (0.0055) |
| No_Low | 0.0514 (0.0032) | 0.0451 (0.0032) | 0.3342 (0.0309) | 0.3194 (0.0304) | 0.0610 (0.0052) |
| No_Volume | 0.0509 (0.0031) | 0.0454 (0.0036) | 0.3346 (0.0303) | 0.3133 (0.0320) | 0.0611 (0.0051) |
| No_Turnover | 0.0513 (0.0040) | 0.0431 (0.0033) | 0.3222 (0.0331) | 0.3102 (0.0311) | 0.0603 (0.0061) |
| FATE | 0.0521 (0.0032) | 0.0491 (0.0033) | 0.3376 (0.0310) | 0.3192 (0.0276) | 0.0619 (0.0057) |

Table 8: Ablation study on the multi-view graph construction on CSI 1000.

| Method | IC | RankIC | ICIR | RankICIR | ICW |
|---|---|---|---|---|---|
| No_Open | 0.0622 (0.0011) | 0.0621 (0.0010) | 0.4442 (0.0205) | 0.3927 (0.0147) | 0.0731 (0.0019) |
| No_Close | 0.0631 (0.0020) | 0.0629 (0.0020) | 0.4042 (0.0211) | 0.4137 (0.0169) | 0.0657 (0.0018) |
| No_High | 0.0627 (0.0015) | 0.0622 (0.0019) | 0.4351 (0.0205) | 0.4130 (0.0164) | 0.0740 (0.0015) |
| No_Low | 0.0610 (0.0020) | 0.0617 (0.0010) | 0.4024 (0.0220) | 0.4157 (0.0170) | 0.0740 (0.0013) |
| No_Volume | 0.0625 (0.0011) | 0.0634 (0.0013) | 0.4446 (0.0203) | 0.4323 (0.0157) | 0.0691 (0.0015) |
| No_Turnover | 0.0632 (0.0012) | 0.0630 (0.0010) | 0.4391 (0.0220) | 0.4302 (0.0161) | 0.0673 (0.0011) |
| FATE | 0.0668 (0.0013) | 0.0645 (0.0011) | 0.4728 (0.0221) | 0.4254 (0.0165) | 0.0765 (0.0015) |

Further, we investigate whether the temporal encoder alone already accounts for most of the predictive gain. To assess this, we conduct an ablation study where the graph module is entirely removed and the model performs prediction solely based on temporal representations, without constructing or propagating through any stock–stock edges.

Concretely, the variant `FATE_NoG` removes all multi-view graphs and bypasses the graph fusion pipeline in Sec. 4.4; the temporal encoder and the prediction head remain unchanged. This design isolates the contribution of cross-stock dependency modeling and allows us to evaluate whether relational information provides incremental value on top of multi-period temporal features.

Tables 9 report the comparison on CSI 500 (C5) and CSI 1000 (C10). Across all metrics, the full FATE model clearly outperforms its graph-free counterpart. These results indicate that while temporal modeling is indeed a major contributor to performance, the graph module provides consistent and non-trivial improvements by capturing cross-stock co-movement patterns that temporal encoders alone cannot model.

Table 9: Effect of removing the graph module on CSI 500 (C5) and CSI 1000 (C10).

| Method | IC | RankIC | ICIR | RankICIR | ICW |
|---|---|---|---|---|---|
| C5_FATE | 0.0521 (0.0032) | 0.0491 (0.0033) | 0.3376 (0.0310) | 0.3192 (0.0276) | 0.0619 (0.0057) |
| C5_NoG | 0.0420 (0.0037) | 0.0335 (0.0034) | 0.2389 (0.0305) | 0.2295 (0.0230) | 0.0499 (0.0052) |
| C10_FATE | 0.0668 (0.0013) | 0.0645 (0.0011) | 0.4728 (0.0221) | 0.4254 (0.0165) | 0.0765 (0.0015) |
| C10_NoG | 0.0604 (0.0012) | 0.0589 (0.0010) | 0.4118 (0.0213) | 0.3781 (0.0210) | 0.0689 (0.0031) |

Overall, these findings confirm that the graph module plays a meaningful and complementary role in FATE, providing relational structure that enhances predictive accuracy beyond temporal modeling alone.

## C.4 EFFECT OF THE STOCK CORRELATION METRIC

This section provides additional analysis on the design of the stock correlation matrix introduced in Sec. 4.4. Instead of adopting a fixed symmetric similarity measure (e.g., cosine similarity or Pearson correlation), FATE constructs an asymmetric co-movement matrix through an attention-style formulation. Given stock embeddings $H_{\text{emb}} \in \mathbb{R}^{N \times F}$, we compute

$$Q = H_{\text{emb}} W_Q, \qquad K = H_{\text{emb}} W_K, \qquad T = \text{sigmoid}(QK^\top) \in \mathbb{R}^{N \times N},$$

where $T_{ij}$ measures how strongly stock $i$ attends to stock $j$. This mechanism produces $T_{ij} \neq T_{ji}$ in general, enabling the model to represent directional influence patterns that naturally arise in financial markets—for instance, small-cap stocks often follow the dynamics of large market leaders, while the reverse relationship may not hold. In contrast, cosine similarity yields a symmetric matrix and does not distinguish the distinct query–key roles between two stocks, limiting its ability to express non-reciprocal relationships or context-dependent interactions. The attention-based formulation also remains numerically stable during training: the stock embeddings are generated through an MLP followed by LayerNorm, keeping their magnitudes well bounded (approximately within $[-8, 8]$ for hidden dimension 64 in our implementation), and the sigmoid function provides smooth gradients across this range. To assess whether this directional scoring provides empirical benefits beyond a simpler symmetric alternative, we replace the attention-based matrix with a cosine-similarity graph and compare the two constructions on both CSI 500 (C5) and CSI 1000 (C10). As shown in Table 10, the attention-based formulation consistently yields higher IC, RankIC, and IR, demonstrating that asymmetric co-movement captures richer cross-stock dependencies than static geometric similarity.

Table 10: Comparison of attention-based (ab) and cosine-similarity metrics on CSI 500 (C5) and CSI 1000 (C10).

| Method | IC | RankIC | ICIR | RankICIR | ICW |
|---|---|---|---|---|---|
| C5_ab | 0.0521 (0.0032) | 0.0491 (0.0033) | 0.3376 (0.0310) | 0.3192 (0.0276) | 0.0619 (0.0057) |
| C5_cosine | 0.0499 (0.0049) | 0.0435 (0.0054) | 0.2898 (0.0325) | 0.2935 (0.0330) | 0.0577 (0.0067) |
| C10_ab | 0.0668 (0.0013) | 0.0645 (0.0011) | 0.4728 (0.0221) | 0.4254 (0.0165) | 0.0765 (0.0015) |
| C10_cosine | 0.0621 (0.0022) | 0.0610 (0.0020) | 0.4283 (0.0253) | 0.3888 (0.0184) | 0.0702 (0.0033) |

Overall, these results validate the use of a asymmetric co-movement metric in FATE, showing both modeling advantages and stable optimization behavior.

## C.5 Effect of the Gating Activation Function in SuperAttention

The gating mechanism in SuperAttention is designed to modulate feature flow by controlling the extent to which temporal and relational information is emphasized or suppressed. This operation requires an activation function capable of expressing both positive and negative modulation. The `tanh` activation naturally satisfies this requirement through its symmetric range $(-1, 1)$, enabling the gate to amplify or attenuate features in a signed manner. In contrast, activations such as `sigmoid` restrict the output to $(0, 1)$, limiting the gate to purely multiplicative attenuation without directional effects.

To examine the practical impact of this choice, we conduct a controlled ablation study replacing the `tanh` gate with a `sigmoid` gate. As shown in Table 11, using `tanh` results in consistently stronger IC and ICIR performance on both CSI 500 and CSI 1000. Although the performance difference is moderate, the `tanh` gate provides a more expressive modulation mechanism and achieves the best overall results. Consequently, we adopt `tanh` as the default activation function for SuperAttention.

Table 11: Effect of gating activation function on CSI 500 (C5) and CSI 1000 (C10).

| Method | IC | RankIC | ICIR | RankICIR | ICW |
|---|---|---|---|---|---|
| C5_tanh | 0.0521 (0.0032) | 0.0491 (0.0033) | 0.3376 (0.0310) | 0.3192 (0.0276) | 0.0619 (0.0057) |
| C5_sigmoid | 0.0513 (0.0037) | 0.0480 (0.0034) | 0.3232 (0.0311) | 0.3190 (0.0310) | 0.0610 (0.0052) |
| C10_tanh | 0.0668 (0.0013) | 0.0645 (0.0011) | 0.4728 (0.0221) | 0.4254 (0.0165) | 0.0765 (0.0015) |
| C10_sigmoid | 0.0651 (0.0019) | 0.0630 (0.0014) | 0.4316 (0.0230) | 0.4223 (0.0200) | 0.0699 (0.0017) |

## C.6 Parameter Sensitivity Analysis

**Effect of the Number of Attention Heads.** We further examine the impact of the number of attention heads. As shown in Tables 12 and 13, using 4 heads achieves the best overall IC, RankIC, and IR metrics on both CSI 500 and CSI 1000. Therefore, we adopt 4 heads as the default choice in our model based on cross-validation results.

Table 12: Sensitivity to the number of attention heads on CSI 500.

| Method | IC | RankIC | ICIR | RankICIR | ICW |
|---|---|---|---|---|---|
| Heads_2 | 0.0507 (0.0030) | 0.0458 (0.0033) | 0.3042 (0.0305) | 0.2997 (0.0270) | 0.0600 (0.0052) |
| Heads_4 | 0.0521 (0.0032) | 0.0491 (0.0033) | 0.3376 (0.0310) | 0.3192 (0.0276) | 0.0619 (0.0057) |
| Heads_8 | 0.0515 (0.0032) | 0.0474 (0.0031) | 0.3346 (0.0313) | 0.3133 (0.0270) | 0.0620 (0.0050) |
| Heads_16 | 0.0498 (0.0044) | 0.0431 (0.0049) | 0.3022 (0.0330) | 0.3002 (0.0311) | 0.0583 (0.0061) |

Table 13: Sensitivity to the number of attention heads on CSI 1000.

| Method | IC | RankIC | ICIR | RankICIR | ICW |
|---|---|---|---|---|---|
| Heads_2 | 0.0647 (0.0015) | 0.0633 (0.0013) | 0.4042 (0.0205) | 0.4200 (0.0174) | 0.0740 (0.0013) |
| Heads_4 | 0.0668 (0.0013) | 0.0645 (0.0011) | 0.4728 (0.0221) | 0.4254 (0.0165) | 0.0765 (0.0015) |
| Heads_8 | 0.0642 (0.0012) | 0.0630 (0.0011) | 0.4457 (0.0213) | 0.4133 (0.0170) | 0.0744 (0.0012) |
| Heads_16 | 0.0630 (0.0020) | 0.0613 (0.0019) | 0.4023 (0.0230) | 0.4002 (0.0196) | 0.0701 (0.0021) |

**Effect of Temporal Partitioning.** We examine how different temporal partitioning schemes affect model performance. For a fixed window length of $T = 60$, we compare the default **T4**: $(T/4, T/2, T)$ split with an alternative **T3**: $(T/3, 2T/3, T)$ split. As reported in Table 14, the two configurations yield similar results on both CSI 500 and CSI 1000, with the default **T4** partition achieving slightly higher IC and ICIR. We therefore adopt **T4** as the default setting in our implementation.

Table 14: Sensitivity to temporal partitioning on CSI 500 (C5) and CSI 1000 (C10).

| Method | IC | RankIC | ICIR | RankICIR | ICW |
|---|---|---|---|---|---|
| C5_T4 | 0.0521 (0.0032) | 0.0491 (0.0033) | 0.3376 (0.0310) | 0.3192 (0.0276) | 0.0619 (0.0057) |
| C5_T3 | 0.0510 (0.0031) | 0.0487 (0.0030) | 0.3242 (0.0311) | 0.3190 (0.0300) | 0.0611 (0.0052) |
| C10_T4 | 0.0668 (0.0013) | 0.0645 (0.0011) | 0.4728 (0.0221) | 0.4254 (0.0165) | 0.0765 (0.0015) |
| C10_T3 | 0.0660 (0.0011) | 0.0651 (0.0014) | 0.4346 (0.0203) | 0.4203 (0.0160) | 0.0697 (0.0012) |

**Effect of Graph Threshold $\tau$.** We investigate how different binarization thresholds $\tau$ influence model performance. As summarized in Tables 15 and 16, $\tau = 0.70$ consistently attains the strongest or near-strongest IC and ICIR metrics across both datasets, while maintaining competitive RankIC and ICW values. Based on these empirical trends, we adopt $\tau = 0.70$ as the default threshold in our implementation.

Table 15: Sensitivity to threshold $\tau$ on CSI 500.

| $\tau$ | IC | RankIC | ICIR | RankICIR | ICW |
|---|---|---|---|---|---|
| 0.50 | 0.0470 (0.0050) | 0.0458 (0.0040) | 0.3042 (0.0335) | 0.2997 (0.0370) | 0.0600 (0.0066) |
| 0.55 | 0.0479 (0.0030) | 0.0458 (0.0040) | 0.3042 (0.0330) | 0.2997 (0.0303) | 0.0600 (0.0059) |
| 0.60 | 0.0500 (0.0033) | 0.0458 (0.0033) | 0.3042 (0.0315) | 0.2997 (0.0260) | 0.0600 (0.0055) |
| 0.65 | 0.0520 (0.0030) | 0.0477 (0.0032) | 0.3242 (0.0307) | 0.3197 (0.0274) | 0.0610 (0.0052) |
| 0.70 | 0.0521 (0.0032) | 0.0491 (0.0033) | 0.3376 (0.0276) | 0.3192 (0.0276) | 0.0619 (0.0057) |
| 0.75 | 0.0505 (0.0031) | 0.0474 (0.0036) | 0.3346 (0.0313) | 0.3133 (0.0300) | 0.0611 (0.0052) |
| 0.80 | 0.0463 (0.0047) | 0.0431 (0.0049) | 0.3022 (0.0331) | 0.3002 (0.0351) | 0.0583 (0.0061) |

Table 16: Sensitivity to threshold $\tau$ on CSI 1000.

| $\tau$ | IC | RankIC | ICIR | RankICIR | ICW |
|---|---|---|---|---|---|
| 0.50 | 0.0628 (0.0031) | 0.0601 (0.0027) | 0.3942 (0.0305) | 0.3907 (0.0207) | 0.0687 (0.0026) |
| 0.55 | 0.0639 (0.0026) | 0.0599 (0.0027) | 0.4042 (0.0291) | 0.4137 (0.0179) | 0.0697 (0.0018) |
| 0.60 | 0.0647 (0.0015) | 0.0628 (0.0019) | 0.4051 (0.0245) | 0.4100 (0.0164) | 0.0740 (0.0015) |
| 0.65 | 0.0660 (0.0015) | 0.0647 (0.0020) | 0.4242 (0.0207) | 0.4197 (0.0174) | 0.0750 (0.0014) |
| 0.70 | 0.0668 (0.0013) | 0.0645 (0.0011) | 0.4728 (0.0221) | 0.4254 (0.0165) | 0.0765 (0.0015) |
| 0.75 | 0.0665 (0.0015) | 0.0654 (0.0013) | 0.4346 (0.0213) | 0.4123 (0.0140) | 0.0691 (0.0013) |
| 0.80 | 0.0630 (0.0010) | 0.0631 (0.0011) | 0.3992 (0.0211) | 0.4002 (0.0151) | 0.0603 (0.0011) |

## C.7 COMPUTATIONAL EFFICIENCY AND SCALABILITY

FATE incorporates multi-view dynamic graph construction and multi-period temporal encoding, which raises natural concerns regarding computational overhead and scalability. We provide both complexity analysis and empirical measurements to assess its feasibility in practical quantitative modeling workflows.

**Complexity.** The dynamic graph construction step computes $K$ correlation matrices per day with complexity $O(KN^2)$. This operation is performed once during preprocessing and is fully vectorized on GPUs, making its cost negligible relative to end-to-end training. During training and inference, the dominant cost arises from attention layers with complexity $O(Nd)$ per layer, comparable to existing graph- and Transformer-based architectures.

**Training and Inference Cost.** Table 17 summarizes end-to-end efficiency on CSI 500 (C5) and CSI 1000 (C10). FATE requires 38.1 s/epoch on C10 and 33.4 s/epoch on C5, which is comparable to THGNN, FactorGCL, and MASTER, while being significantly faster than heavier architectures such as StockMixer, TSRM, HATS, or FactorVAE. For inference, FATE achieves 2.39 s/day on C10 and 1.74 s/day on C5, satisfying the latency requirements for daily quantitative trading pipelines.

Table 17: Overall computational cost comparison: training time (per epoch), inference latency (per day), and peak memory usage on CSI 1000 (C10) and CSI 500 (C5).

| Method | Train Time (s/epoch) | | Inference (s/day) | | Memory (MB) | |
|---|---|---|---|---|---|---|
| | C10 | C5 | C10 | C5 | C10 | C5 |
| MLP | 4.98 | 5.01 | 0.89 | 0.94 | 822 | 820 |
| GRU | 6.44 | 5.93 | 0.97 | 0.92 | 1590 | 1068 |
| LSTM | 6.54 | 6.11 | 0.99 | 0.93 | 1618 | 1068 |
| Transformer | 17.68 | 17.27 | 1.83 | 1.36 | 1565 | 1214 |
| HIST | 22.11 | 18.03 | 3.88 | 2.87 | 1688 | 1314 |
| FactorVAE | 43.21 | 43.04 | 4.25 | 4.20 | 1580 | 1252 |
| THGNN | 16.78 | 16.12 | 1.53 | 1.40 | 2324 | 1470 |
| StockMixer | 52.75 | 52.58 | 1.97 | 1.93 | 1854 | 1316 |
| HATS | 27.50 | 24.78 | 2.18 | 1.83 | 1798 | 1340 |
| FactorGCL | 20.54 | 15.53 | 3.76 | 2.85 | 1688 | 1306 |
| TSRM | 48.58 | 36.82 | 10.88 | 8.04 | 1666 | 1116 |
| MASTER | 20.00 | 19.77 | 3.52 | 3.50 | 2202 | 2103 |
| FATE | 38.09 | 33.45 | 2.39 | 1.73 | 3522 | 1766 |

**Memory Usage.** The GPU memory footprint is $3.52\,\text{GB}$ on C10 and $1.77\,\text{GB}$ on C5. This cost reflects the storage of multi-view graphs and multi-period temporal embeddings. Although higher than RNN-based baselines, the memory overhead remains well within the limits of commonly used 24–32 GB GPUs, and scales linearly with the number of stocks.

**Scalability.** The transition from C5 to C10 results in a smooth and predictable increase in runtime and memory, demonstrating the model's ability to scale with universe size. Since the most expensive step—graph construction—is amortized as a one-time preprocessing computation, and the dominant per-layer operations scale as $O(Nd)$, FATE remains readily applicable to larger universes (e.g., 3k–4k tradable A-share stocks).

Overall, FATE exhibits computational cost comparable to existing attention- and graph-based methods, while maintaining practical training and inference speed suitable for real-world deployment.

### C.8 VISUALIZATION OF SUPERATTENTION WEIGHTS

To examine the effect of the self-supervised graph attention loss, we visualize the SuperAttention matrices for a random subset of 50 stocks. Given the sparse nature of stock interactions, this subset provides a clearer view of how attention is allocated across stocks. For both settings (with and without the graph-regularized loss), we apply row-wise softmax normalization and plot the resulting attention heatmaps.

Figure 3 shows that the graph-regularized loss produces noticeably sharper distributions. The standard deviation of attention coefficients increases from $0.016$ (without loss) to $0.021$ (with loss), indicating stronger contrast between important and non-important connections. This aligns with the expected sparsity of financial dependencies and demonstrates that the attention loss encourages more selective, interpretable, and economically meaningful co-movement patterns.

## D USE OF LARGE LANGUAGE MODELS

Large language models (LLMs) were employed exclusively as general-purpose writing assistants to refine grammar, improve clarity, and polish the presentation of the manuscript. They were not used for research ideation, methodology development, experimental design, analysis, or drawing conclusions. All technical contributions, experiments, and results in this work were conceived, implemented, and validated entirely by the authors. This usage does not affect the scientific validity or originality of the work.

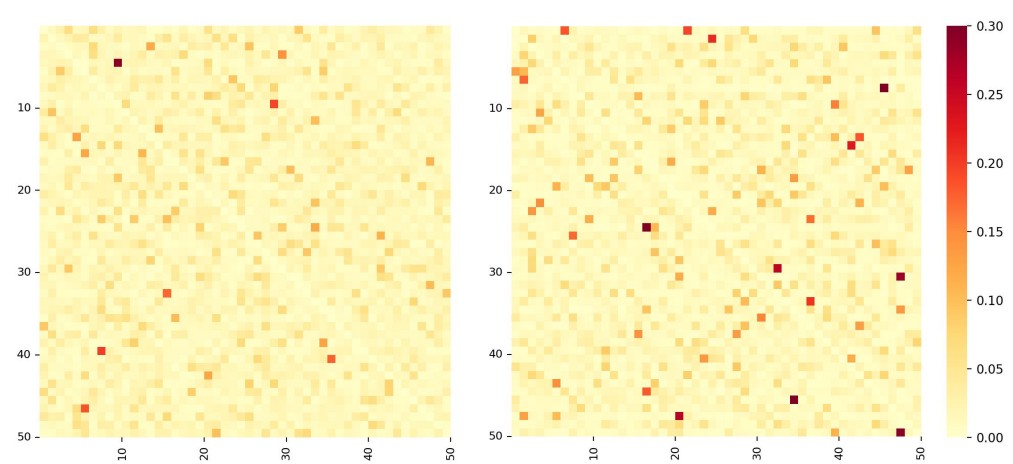

Figure 3: Visualization of SuperAttention weights for 50 sampled stocks. Left: without the graph-regularized loss. Right: with the graph-regularized loss.

