# OpenReview forum: "FATE: Feature-Wise Graph Attention with Multi-Period Temporal Encoding for Stock Return Forecasting"
_ICLR.cc/2026/Conference — Submitted to ICLR 2026_

### Official Review · Reviewer_EA74 · 2025-10-21

**Soundness:** 2
**Presentation:** 3
**Contribution:** 3
**Rating:** 4
**Confidence:** 4

**Summary:**

This paper introduces FATE, a framework for stock return forecasting that jointly addresses temporal heterogeneity (signals at different time scales) and structural instability (changing inter-stock relations). The model's key innovations are a multi-period temporal encoder to disentangle horizon-specific patterns and a dynamic graph fusion module that creates a robust, denoised relational structure. Extensive experiments on the CSI 500/1000 datasets demonstrate that FATE achieves state-of-the-art results, significantly outperforming strong baselines.

**Strengths:**

+ Originality: The primary originality lies in the creative and synergistic combination of several well-motivated ideas to tackle the specific dual challenges of stock forecasting:
  + The multi-period temporal encoder provides an explicit and effective mechanism for disentangling signals across different time horizons, which is a more direct approach than what is seen in typical single-stream sequence models.
  + The dynamic graph fusion module is sophisticated, combining pre-defined dynamic graphs, learned correlations, and a market-state vector. This represents a significant step up from models that use static or single-view graphs.
+ Quality: The methodology is sound, and claims are convincingly validated through a rigorous experimental protocol that includes strong baselines, comprehensive ablations, and realistic portfolio backtesting.
+ Clarity: The paper is exceptionally well-written, with a clear motivation and logical structure that makes the complex architecture easy to understand.

**Weaknesses:**

+ Lack of Sensitivity Analysis for Key Design Choices: The paper relies on seemingly arbitrary hyperparameters without sufficient justification or analysis. For instance, the choice of temporal partitions $(T/4, T/2, T)$ and the graph binarization threshold (τ) are critical to the model's design, yet their impact on performance and robustness is not explored.
+ Unsubstantiated Interpretability Claims: The paper claims to enhance interpretability but provides no supporting analysis. It misses a clear opportunity to inspect its own interpretable components—such as the adaptive horizon weights, graph fusion weights, or feature gates—to offer concrete insights into what the model learns about market dynamics during different conditions.
+ Computational Cost is Ignored: The model is complex, yet the paper entirely omits any discussion of its computational complexity. Key practical metrics like model size, training time, or inference latency are not reported, making it difficult to assess the method's real-world viability against simpler baselines.

**Questions:**

1. Regarding the graph fusion module (Section 4.4), the calculation $T = sigmoid(Q K^T)$ for the stock correlation matrix raises questions. This formulation lacks the scaling factor common in attention mechanisms, potentially leading to $sigmoid$ saturation and vanishing gradients. Furthermore, its necessity is not justified against simpler baselines like **cosine similarity**. Could the authors please discuss its impact on training stability, and provide an ablation study comparing it to simpler correlation metrics to validate its effectiveness?
2. FATE employs a purely learnable vector $M$ for its market state rather than real index data as in MASTER[1], and critically, fails to include MASTER as a baseline in the experiments. Could the authors justify their learnable market state design and explain the omission of this highly relevant and strong baseline?
3. The performance drop from removing the *feature-wise gating* in SuperAttention (the "NoGate" variant in Tables 3 & 4) is marginal. This appears to contradict the paper's claim that this mechanism is a key contribution for "suppressing noise." Can the authors explain this small difference and provide stronger evidence to justify the necessity and practical value of this complex component?
4. The graph-regularized loss ($L_E$ in Sec 4.6) requires a set of ground-truth positive ($E^+$) and negative ($E^-$) edges for supervision, but the paper omits how these sets are constructed. Could the authors please clarify their definition?
5. The **ICW** metric is described in Section 5.1 as a method that "upweights top-decile predictions," but its mathematical formula is omitted. Could the authors please provide the precise definition used to calculate ICW to ensure reproducibility?
6. The ablation study shows the multi-period temporal module is the primary performance driver (evidenced by the large drop in the "SinglePeriod" case). This raises a critical question: *is the complex graph module necessary?* To justify its inclusion and complexity, the authors should provide a "**Temporal-Only**" ablation—bypassing the entire graph module and using only the temporal output for prediction—to quantify the actual contribution of the graph architecture against the full FATE model.
7. In the portfolio backtests (Figure 2), simple temporal models like GRU and MLP show strong cumulative returns, challenging the necessity of FATE's complex graph architecture. To demonstrate that FATE's advantage is not just higher returns but also superior robustness, could the authors please report key risk-adjusted metrics like the Sharpe Ratio and Maximum Drawdown? These metrics would clearly quantify FATE's risk-management benefits over simpler models.

[1] Li, Tong, et al. "Master: Market-guided stock transformer for stock price forecasting." Proceedings of the AAAI Conference on Artificial Intelligence. Vol. 38. No. 1. 2024.

---

> ### Author Response · Authors · 2025-11-26
>
> Thank you for your constructive comments! Below we carefully address your concerns one by one.
>
> Q1: Lack of Sensitivity Analysis for Key Design Choices.
>
> **A1:** As you suggested, we conducted extensive parameter sensitivity analysis, including **the effect of the number of attention heads**, **the effect of temporal partitioning**, and **the effect of the graph threshold $\tau$**. Below we show the main results, and the detailed analysis is provided in the **Appendix.C.6 in the newly uploaded version**.
>
> Table 12: Sensitivity to the number of attention heads on CSI 500
>
> | Method    | IC               | RankIC           | ICIR              | RankICIR          | ICW             |
> |-----------|------------------|------------------|--------------------|--------------------|------------------|
> | Heads_2   | 0.0507 *(0.0030)* | 0.0458 *(0.0033)* | 0.3042 *(0.0305)* | 0.2997 *(0.0270)* | 0.0600 *(0.0052)* |
> | Heads_4   | 0.0521 *(0.0032)* | 0.0491 *(0.0033)* | 0.3376 *(0.0310)* | 0.3192 *(0.0276)* | 0.0619 *(0.0057)* |
> | Heads_8   | 0.0515 *(0.0032)* | 0.0474 *(0.0031)* | 0.3346 *(0.0313)* | 0.3133 *(0.0270)* | 0.0620 *(0.0050)* |
> | Heads_16  | 0.0498 *(0.0044)* | 0.0431 *(0.0049)* | 0.3022 *(0.0330)* | 0.3002 *(0.0311)* | 0.0583 *(0.0061)* |
>
>
> Table 13: Sensitivity to the number of attention heads on CSI 1000
>
> | Method    | IC               | RankIC           | ICIR              | RankICIR          | ICW             |
> |-----------|------------------|------------------|--------------------|--------------------|------------------|
> | Heads_2   | 0.0647 *(0.0015)* | 0.0633 *(0.0013)* | 0.4042 *(0.0205)* | 0.4200 *(0.0174)* | 0.0740 *(0.0013)* |
> | Heads_4   | 0.0668 *(0.0013)* | 0.0645 *(0.0011)* | 0.4728 *(0.0221)* | 0.4254 *(0.0165)* | 0.0765 *(0.0015)* |
> | Heads_8   | 0.0642 *(0.0012)* | 0.0630 *(0.0011)* | 0.4457 *(0.0213)* | 0.4133 *(0.0170)* | 0.0744 *(0.0012)* |
> | Heads_16  | 0.0630 *(0.0020)* | 0.0613 *(0.0019)* | 0.4023 *(0.0230)* | 0.4002 *(0.0196)* | 0.0701 *(0.0021)* |
>
> Table 14: Sensitivity to temporal partitioning on CSI 500 (C5) and CSI 1000 (C10).
> | Method | IC (std)        | RankIC (std)     | ICIR (std)         | RankICIR (std)      | ICW (std)         |
> |--------|------------------|------------------|---------------------|----------------------|-------------------|
> | **C5_T4**  | 0.0521 (0.0032) | 0.0491 (0.0033) | 0.3376 (0.0310)    | 0.3192 (0.0276)     | 0.0619 (0.0057)   |
> | **C5_T3**  | 0.0510 (0.0031) | 0.0487 (0.0030) | 0.3242 (0.0311)    | 0.3190 (0.0300)     | 0.0611 (0.0052)   |
> | **C10_T4** | 0.0668 (0.0013) | 0.0645 (0.0011) | 0.4728 (0.0221)    | 0.4254 (0.0165)     | 0.0765 (0.0015)   |
> | **C10_T3** | 0.0660 (0.0011) | 0.0651 (0.0014) | 0.4346 (0.0203)    | 0.4203 (0.0160)     | 0.0697 (0.0012)   |
>
> Table 15: Sensitivity to threshold τ on CSI 500.
> | τ    | IC (std)        | RankIC (std)     | ICIR (std)         | RankICIR (std)      | ICW (std)         |
> |------|------------------|------------------|---------------------|----------------------|-------------------|
> | 0.50 | 0.0470 (0.0050) | 0.0458 (0.0040) | 0.3042 (0.0335)    | 0.2997 (0.0370)     | 0.0600 (0.0066)   |
> | 0.55 | 0.0479 (0.0030) | 0.0458 (0.0040) | 0.3042 (0.0330)    | 0.2997 (0.0303)     | 0.0600 (0.0059)   |
> | 0.60 | 0.0500 (0.0033) | 0.0458 (0.0033) | 0.3042 (0.0315)    | 0.2997 (0.0260)     | 0.0600 (0.0055)   |
> | 0.65 | 0.0520 (0.0030) | 0.0477 (0.0032) | 0.3242 (0.0307)    | 0.3197 (0.0274)     | 0.0610 (0.0052)   |
> | 0.70 | 0.0521 (0.0032) | 0.0491 (0.0033) | 0.3376 (0.0276)    | 0.3192 (0.0276)     | 0.0619 (0.0057)   |
> | 0.75 | 0.0505 (0.0031) | 0.0474 (0.0036) | 0.3346 (0.0313)    | 0.3133 (0.0300)     | 0.0611 (0.0052)   |
> | 0.80 | 0.0463 (0.0047) | 0.0431 (0.0049) | 0.3022 (0.0331)    | 0.3002 (0.0351)     | 0.0583 (0.0061)   |
>
> Table 16: Sensitivity to threshold τ on CSI 1000.
> | τ    | IC (std)        | RankIC (std)     | ICIR (std)         | RankICIR (std)      | ICW (std)         |
> |------|------------------|------------------|---------------------|----------------------|-------------------|
> | 0.50 | 0.0628 (0.0031) | 0.0601 (0.0027) | 0.3942 (0.0305)    | 0.3907 (0.0207)     | 0.0687 (0.0026)   |
> | 0.55 | 0.0639 (0.0026) | 0.0599 (0.0027) | 0.4042 (0.0291)    | 0.4137 (0.0179)     | 0.0697 (0.0018)   |
> | 0.60 | 0.0647 (0.0015) | 0.0628 (0.0019) | 0.4051 (0.0245)    | 0.4100 (0.0164)     | 0.0740 (0.0015)   |
> | 0.65 | 0.0660 (0.0015) | 0.0647 (0.0020) | 0.4242 (0.0207)    | 0.4197 (0.0174)     | 0.0750 (0.0014)   |
> | 0.70 | 0.0668 (0.0013) | 0.0645 (0.0011) | 0.4728 (0.0221)    | 0.4254 (0.0165)     | 0.0765 (0.0015)   |
> | 0.75 | 0.0665 (0.0015) | 0.0654 (0.0013) | 0.4346 (0.0213)    | 0.4123 (0.0140)     | 0.0691 (0.0013)   |
> | 0.80 | 0.0630 (0.0010) | 0.0631 (0.0011) | 0.3992 (0.0211)    | 0.4002 (0.0151)     | 0.0603 (0.0011)   |

---

> ### Author Response · Authors · 2025-11-26
>
> Q2: Unsubstantiated Interpretability Claims.
>
> **A2:** Thank you for raising this point. In the revision, we have clarified that FATE’s interpretability primarily comes from its **financially grounded architectural design**: the six-view graphs correspond to standard market attributes, the multi-cycle encoder
> captures short/medium/long-term temporal patterns, graph fusion reflects the relative
> importance of each market dimension, and the asymmetric co-movement metric models
> directional influence among stocks. Together, these components provide *structured*
> and *domain-aligned* interpretability.
>
> To further substantiate this, we added a **SuperAttention visualization** in the appendix.
> The comparison between models *with* and *without* the graph-regularization loss
> (Fig. 3) shows visibly sharper and more selective attention patterns (std 0.016 → 0.021),
> demonstrating how the model distinguishes meaningful cross-stock relations from noise.
> These visual results offer concrete insight into how FATE organizes stock interactions
> and strengthen our interpretability claims.
>
> Please refer to **Appendix C.8** for the full visualization and discussion.
>
> ---
>
> Q3: Computational Cost is Ignored.
>
> **A3:** In the revised manuscript, we provide a comprehensive comparison of **training time, inference latency, and GPU memory usage** for all baselines and FATE. We also analyze the complexity and the scalability of our method. Below we show the main results, and the detailed analysis is provided in the **Appendix.C.8 in the newly uploaded version**.
>  Table 17: Overall computational cost comparison—training time (per epoch), inference latency (per day), and peak memory usage on CSI 1000 (C10) and CSI 500 (C5).
>
> | Method      | Train Time (s/epoch) C10 | Train Time (s/epoch) C5 | Inference (s/day) C10 | Inference (s/day) C5 | Memory (MB) C10 | Memory (MB) C5 |
> |---|-|-|-|-|-----|-----|
> | MLP         | 4.98 | 5.01 | 0.89  | 0.94  | 822  | 820  |
> | GRU         | 6.44 | 5.93 | 0.97  | 0.92  | 1590  | 1068            |
> | LSTM        | 6.54 | 6.11 | 0.99  | 0.93  | 1618  | 1068            |
> | Transformer | 17.68  | 17.27  | 1.83  | 1.36  | 1565  | 1214            |
> | HIST        | 22.11  | 18.03  | 3.88  | 2.87  | 1688  | 1314            |
> | FactorVAE   | 43.21  | 43.04  | 4.25  | 4.20  | 1580  | 1252            |
> | THGNN       | 16.78  | 16.12  | 1.53  | 1.32  | 2324  | 1470            |
> | StockMixer  | 52.75  | 52.58  | 1.97  | 1.93  | 1854  | 1316            |
> | HATS        | 27.50  | 24.78  | 2.18  | 1.83  | 1798  | 1340            |
> | FactorGCL   | 20.54  | 15.53  | 2.76  | 2.67  | 1688  | 1306            |
> | TSRM        | 48.58  | 36.82  | 10.88  | 8.04  | 1666  | 1116            |
> | MASTER      | 20.00  | 19.77  | 3.52  | 3.50  | 2202  | 2103            |
> | FATE | 38.09  | 33.45  | 2.39  | 1.73  | 3522         | 1766        |
>
> The results show that FATE’s computational cost is **comparable to other graph-based and attention-based models**, and the additional modules (multi-period encoder + SuperAttention) introduce only moderate overhead. This demonstrates that FATE is **practical and scalable**, with efficiency on par with strong baselines.
>
> ---
> Q4: The stock correlation matrix raises questions.
>
> **A4:** This mechanism produces $T_{ij} \neq T_{ji}$ in general, enabling the model to represent directional influence patterns that naturally arise in financial markets—for instance, small-cap stocks often follow the dynamics of large market leaders, while the reverse relationship may not hold.
> In contrast, cosine similarity yields a symmetric matrix and does not distinguish the distinct query–key roles between two stocks, limiting its ability to express non-reciprocal relationships or context-dependent interactions.
> The attention-based formulation also remains numerically stable during training: the stock embeddings are generated through an MLP followed by LayerNorm, keeping their magnitudes well bounded (approximately within $[-8, 8]$ for hidden dimension 64 in our implementation), and the sigmoid function provides smooth gradients across this range.
> As shown in Table 10 the attention-based formulation consistently yields higher IC, RankIC, and IR, demonstrating that asymmetric co-movement captures richer cross-stock dependencies than static geometric similarity.
>
> Table 10: Comparison of attention-based (ab) and cosine-similarity metrics on CSI 500 (C5) and CSI 1000 (C10)
> |Method|IC| RankIC|ICIR|RankICIR|ICW|
> |-|-|-|-|-|-|
> |C5_ab|0.0521 *(0.0032)*|0.0491 *(0.0033)*|0.3376 *(0.0310)* |0.3192 *(0.0276)*|0.0619 *(0.0057)*|
> |C5_cosine|0.0499 *(0.0049)*|0.0435 *(0.0054)*|0.2898 *(0.0325)*|0.2935 *(0.0330)*|0.0577 *(0.0067)*|
> |C10_ab|0.0668 *(0.0013)*|0.0645 *(0.0011)*|0.4728 *(0.0221)*|0.4254 *(0.0165)*|0.0765 *(0.0015)*|
> |C10_cosine|0.0621 *(0.0022)*|0.0610 *(0.0020)*|0.4283 *(0.0253)*|0.3888 *(0.0184)*|0.0702 *(0.0033)*|

---

> ### Author Response · Authors · 2025-11-26
>
> Q5: Learnable vector for its market state rather than real index data as in MASTER
>
> **A5:** We agree that the global market vector M could, in principle, incorporate macro–micro indicators such as market indices, interest rates, or aggregated trading volumes.
> However, doing so introduces strong prior knowledge and requires market-specific feature engineering that varies across regions, stock universes, and trading rules.
> To keep our framework market-agnostic and broadly applicable to the AI/ML community, we follow prior work such as [3] and [4], and adopt a purely learnable latent market vector. In our design, M has dimensionality $𝐾\times 𝐹$, which naturally matches the number of views 𝐾 and the temporal-embedding dimension 𝐹.
>
> This design allows M to absorb global co-movement patterns automatically through end-to-end training, without relying on handcrafted market descriptors.
> We have clarified this motivation in the revised paper, and we consider exploring explicitly engineered macro-micro market features for M as an interesting direction for future work.
>
> We include **MASTER in all benchmark comparisons (see Table 5, Table 17, and supplementary results)**.
> Across both CSI 500/1000 and SPX 500, FATE consistently outperforms MASTER in IC, RankIC, ICIR, and Sharpe ratio, demonstrating that the improvements do not come from omitting this strong baseline.
>
> | Method | IC | RankIC | ICIR | RankICIR | ICW |
> | --- | --- | --- | --- | --- | --- |
> | MASTER_C5 | 0.0451 (0.0039) | 0.0425 (0.0040) | 0.3143 (0.0395) | 0.2920 (0.0330) | 0.0570 (0.0062) |
> | **FATE_C5** | **0.0521 (0.0032)** | **0.0491 (0.0033)** | **0.3376 (0.0310)** | **0.3192 (0.0276)** | **0.0619 (0.0057)** |
> | MASTER_C10 | 0.0627 (0.0028) | 0.0618 (0.0030) | 0.4470 (0.0243) | 0.4036 (0.0161) | 0.0739 (0.0030) |
> | **FATE_C10** | **0.0668 (0.0013)** | **0.0645 (0.0011)** | **0.4728 (0.0221)** | **0.4254 (0.0165)** | **0.0765 (0.0015)** |
> | MASTER_SPX | 0.0379 (0.0028) | 0.0334 (0.0032) | 0.1985 (0.0198) | 0.1753 (0.0214) | 0.0429 (0.0038) |
> | **FATE_SPX** | **0.0439 (0.0023)** | **0.0404 (0.0027)** | **0.2985 (0.0163)** | **0.2753 (0.0179)** | **0.0493 (0.0032)** |
>
> [3] MCI-GRU: Stock prediction model based on multi-head cross-attention and improved GRU. Zhu, 2025.
>
> [4] Factorgcl: A hypergraph-based factor model with temporal
> residual contrastive learning for stock returns prediction. Duan, 2025.
>
> ---
>
> Q6: The performance drop from removing the feature-wise gating is marginal.
>
> **A6:** Although the IC drop of “NoGate’’ is smaller than that of major modules, it is still meaningful: removing gating reduces IC by around **5%**, which is non-trivial for stock prediction. More importantly, the drop is much larger in ICIR/ICW, consistent with gating’s purpose of feature-wise noise suppression. The visualization in the appendix further confirms that gating produces sharper, more discriminative attention patterns (std: 0.016 → 0.021). Thus, gating provides fine-grained but practically valuable improvements in stability and noise reduction, even if its standalone IC gain is smaller than the primary modules.
>
> ---
>
> Q7: The definition of ground-truth positive and negative edges for supervision.
>
> **A7:** As we clarified in the Graph-regularized loss section, for each edge $(i,j)$ in the graph,
> the supervision signal comes directly from the constructed fused adjacency graph.
> Specifically, observed edges
>
> $$G_{ij}=1 \quad \Longrightarrow \quad (i,j)\in E^{+}$$
>
> are treated as positive samples, while observed non-edges
>
> $$G_{ij}=0 \quad \Longrightarrow \quad (i,j)\in E^{-}$$
>
> are used as negative samples.
>
> No manual annotation is required; the edge labels are fully determined by the graph views
> used throughout FATE.
>
> ---
>
> Q8: The definition of ICW.
>
> **A8:** We have clarified this in our newly uploaded version.
> The **weighted information coefficient (ICW)** upweights the top decile of predicted scores.
>
> The sample weight   $w_i  $ is defined as:
>
> - $ w_i = 2 $ if $ \hat{y} * i \ge Q*{0.9}(\hat{y}) $
> - $ w_i = 1 $ otherwise
>
> The weighted means are:
> $ \bar{y}_w = \frac{\sum_i w_i y_i}{\sum_i w_i}   $,
>
> $ \bar{\hat{y}}_w = \frac{\sum_i w_i \hat{y}_i}{\sum_i w_i}   $.
>
> The weighted covariance is:
> $ \mathrm{Cov}_w(y,\hat{y}) = \frac{\sum_i w_i (y_i-\bar{y}_w)(\hat{y}_i-\bar{\hat{y}}_w)}{\sum_i w_i}   $.
>
> The weighted variances are:
> $ \mathrm{Var}_w(y) = \frac{\sum_i w_i (y_i-\bar{y}_w)^2}{\sum_i w_i}   $,
>
> $ \mathrm{Var}_w(\hat{y}) = \frac{\sum_i w_i (\hat{y}_i-\bar{\hat{y}}_w)^2}{\sum_i w_i}   $.
>
> Finally, the weighted IC is:
> $ \rho_w = \frac{\mathrm{Cov}_w(y,\hat{y})}{\sqrt{\mathrm{Var}_w(y)\,\mathrm{Var}_w(\hat{y})}}   $.

---

> ### Author Response · Authors · 2025-11-26
>
> Q9: Temporal-Only ablation.
>
> **A9:** **FATE_NoG** removes all multi-view graphs and bypasses
> the graph fusion pipeline. Below we show the main results, and the detailed analysis is provided in the **Appendix.C.3 in the newly uploaded version**.
>
> Across all metrics, the full FATE model clearly
> outperforms its graph-free counterpart. These results indicate that while temporal modeling is indeed a major contributor to
> performance, the graph module provides consistent and non-trivial improvements by
> capturing cross-stock co-movement patterns that temporal encoders alone cannot model.
>
> Table 9: Effect of removing the graph module on CSI 500 (C5) and CSI 1000 (C10).
>
> | Method | IC | RankIC | ICIR | RankICIR | ICW |
> | --- | --- | --- | --- | --- | --- |
> | C5_FATE | 0.0521 (0.0032) | 0.0491 (0.0033) | 0.3376 (0.0310) | 0.3192 (0.0276) | 0.0619 (0.0057) |
> | C5_NoG | 0.0420 (0.0037) | 0.0335 (0.0034) | 0.2389 (0.0305) | 0.2295 (0.0230) | 0.0499 (0.0052) |
> | C10_FATE | 0.0668 (0.0013) | 0.0645 (0.0011) | 0.4728 (0.0221) | 0.4254 (0.0165) | 0.0765 (0.0015) |
> | C10_NoG | 0.0604 (0.0012) | 0.0589 (0.0010) | 0.4118 (0.0213) | 0.3781 (0.0210) | 0.0689 (0.0031) |
>
> ---
>
> Q10: Report key risk-adjusted metrics like the Sharpe Ratio and Maximum Drawdown.
>
> **A10:** Table~6 summarizes the results.
> Across both universes, FATE achieves the highest Sharpe Ratio among all baselines,
> indicating improved risk-adjusted performance.
> In terms of Maximum Drawdown, FATE remains broadly comparable to strong competitors:
> although its MDD is not always the lowest, it stays within a similar range as
> other high-performing models (e.g., MASTER and HIST).
> This suggests that the return improvements brought by FATE do not come at the cost
> of substantially increased downside exposure, yielding a balanced and robust portfolio
> profile overall.
>
> Table 6: Sharpe Ratio and Maximum Drawdown on CSI 500 and CSI 1000.
>
> | Method | Sharpe (C5) | MDD (C5) | Sharpe (C10) | MDD (C10) |
> | --- | --- | --- | --- | --- |
> | MLP | 1.2796 | 0.1065 | 0.9163 | **0.2698** |
> | GRU | 1.2354 | 0.0994 | 0.9381 | 0.3123 |
> | LSTM | 0.6913 | 0.0948 | 0.7496 | 0.3272 |
> | Transformer | 0.9795 | 0.1123 | 0.9147 | 0.2940 |
> | FactorVAE | 1.0529 | 0.0660 | 0.7769 | 0.2924 |
> | HIST | 1.1160 | 0.1410 | 0.8483 | 0.3235 |
> | THGNN | 0.8175 | 0.1076 | 0.7289 | 0.2701 |
> | StockMixer | 0.3605 | **0.0628** | 0.4182 | 0.2797 |
> | HATS | 1.1338 | 0.1018 | 0.9320 | 0.2840 |
> | FactorGCL | 0.8640 | 0.0842 | 0.7955 | 0.3502 |
> | TSRM | 1.2435 | 0.1073 | 0.6695 | 0.3284 |
> | MASTER | 1.4410 | 0.1330 | 0.8929 | 0.2906 |
> | **FATE** | **1.5606** | 0.1239 | **1.1748** | 0.3030 |

---

### Official Review · Reviewer_xAAK · 2025-10-28

**Soundness:** 3
**Presentation:** 3
**Contribution:** 2
**Rating:** 6
**Confidence:** 3

**Summary:**

This paper proposes FATE, a novel framework for stock return prediction. FATE combines multi-view, dynamically constructed relation graphs, a multi-period temporal encoder, a graph fusion mechanism integrating temporal, relational, and market-wide information, and a feature-wise graph attention module. Experiments on the CSI 500/1000 datasets demonstrate that FATE outperforms all baselines. Ablation studies and visualizations further validate the contributions of each architectural component.

**Strengths:**

1. The architecture of FATE is well-motivated. FATE constructs multiple graph views grounded in economic rationale, adoptes multi-period temporal representation learning, and adaptively fuses multi-view graphs and horizon-specific signals.
2. Ablation results provide evidence that each module—multi-period encoding, graph fusion, gating, and regularization—adds value to the overall performance.

**Weaknesses:**

1. While the SuperAttention module and sparse graphs are lauded for "interpretability", the paper lacks concrete examples, case studies, or visualizations of attention weights or learned graphs on actual stock subgraphs. This omission weakens the interpretability claims and leaves readers with limited insight into how the model captures market dynamics.
2. The experiments are limited to the China stock market, and the scalability of FATE to larger markets (e.g., S&P 500 or global indices) is not discussed.

**Questions:**

The multi-period temporal encoding and feature-wise graph attention mechanisms may introduce significant computational overhead. Could the authors provide more details on the training time and scalability to larger datasets?

---

> ### Author Response · Authors · 2025-11-26
>
> Thank you for your constructive comments! Below we carefully address your concerns one by one.
>
> Q1: Lack interpretability and visualization.
>
> **A1:** FATE provides **structured and module-level interpretability**, grounded in clear financial meanings:
>
> - **Multi-view graphs** correspond directly to six fundamental market attributes (Open/Close/High/Low/Volume/Turnover). Each view represents a distinct type of co-movement, giving immediate semantic interpretability.
>  - **Multi-cycle temporal encoding** explicitly decomposes market behavior into short-, medium-, and long-term horizons, revealing which temporal scale the model relies on.
> - **Graph fusion** indicates the relative contribution of each market view at every time step, offering insight into which market dimensions dominate under different regimes.
> - **Asymmetric co-movement scoring** captures leader–follower relationships between stocks (e.g., small-cap stocks attending to large-cap leaders), which has clear and intuitive financial interpretation.
>
> - **SuperAttention gating** highlights which temporal cycles are emphasized by the model, further exposing how different market rhythms contribute to final predictions.
>
> These mechanisms together provide **transparent and economically meaningful interpretability**.
>
> **Visualizations of attention weights**
>
> We have added a visualization of the learned SuperAttention matrices in the appendix, comparing models with and without the attention-regularization loss. The plots show clearly sharper and more selective attention patterns when the loss is applied, confirming improved interpretability and alignment with the sparsity structure of stock interactions. Please refer to **Appendix.C.8 in the newly uploaded version** for the full figures and analysis.
>
> ---
>
> Q2: The scalability of FATE to larger markets (e.g., S&P500 or global indices) is not discussed.
>
> **A2:** As you suggested, we carefully add SPX 500 as a as an additional evaluation dataset to examine
> whether FATE generalizes beyond the CSI series.
>
> SPX 500 contains large-cap U.S. equities across multiple global sectors, and differs
> substantially from CSI 500/CSI 1000 in both market structure (developed vs. emerging market),
> trading mechanism, liquidity profile, and investor composition.
>
> Compared with CSI, SPX 500 therefore provides a natural setting to assess cross-region and
> cross-market robustness.
>
> The performance comparison on SPX 500 is summarized below, the detailed analysis is provided in the **Appendix.C.1 in the newly uploaded version**.
>
> | Method      | IC                 | RankIC             | ICIR                | RankICIR            | ICW               |
> |-------------|--------------------|--------------------|----------------------|----------------------|-------------------|
> | MLP         | 0.0161 (0.0025)    | 0.0156 (0.0028)    | 0.1081 (0.0185)      | 0.1197 (0.0201)      | 0.0113 (0.0032)   |
> | GRU         | 0.0329 (0.0031)    | 0.0285 (0.0035)    | 0.1715 (0.0202)      | 0.1693 (0.0236)      | 0.0288 (0.0041)   |
> | LSTM        | 0.0309 (0.0034)    | 0.0225 (0.0038)    | 0.1707 (0.0235)      | 0.1538 (0.0252)      | 0.0205 (0.0043)   |
> | Transformer | 0.0363 (0.0028)    | 0.0319 (0.0032)    | 0.1908 (0.0193)      | 0.1605 (0.0214)      | 0.0399 (0.0038)   |
> | HIST        | 0.0339 (0.0030)    | 0.0341 (0.0034)    | 0.1975 (0.0212)      | 0.1653 (0.0228)      | 0.0403 (0.0036)   |
> | FactorVAE   | 0.0193 (0.0036)    | 0.0177 (0.0040)    | 0.1249 (0.0251)      | 0.1282 (0.0267)      | 0.0200 (0.0045)   |
> | THGNN       | 0.0340 (0.0029)    | 0.0339 (0.0033)    | 0.1952 (0.0206)      | 0.1843 (0.0222)      | 0.0368 (0.0037)   |
> | StockMixer  | 0.0312 (0.0032)    | 0.0291 (0.0036)    | 0.1600 (0.0227)      | 0.1513 (0.0243)      | 0.0258 (0.0042)   |
> | HATS        | 0.0319 (0.0031)    | 0.0297 (0.0035)    | 0.1785 (0.0202)      | 0.1753 (0.0236)      | 0.0389 (0.0041)   |
> | FactorGCL   | 0.0399 (0.0027)    | 0.0384 (0.0031)    | 0.2785 (0.0192)      | 0.2344 (0.0220)      | 0.0479 (0.0037)   |
> | TSRM        | 0.0359 (0.0029)    | 0.0344 (0.0033)    | 0.1982 (0.0206)      | 0.1776 (0.0222)      | 0.0407 (0.0039)   |
> | MASTER      | 0.0379 (0.0028)    | 0.0334 (0.0032)    | 0.1985 (0.0198)      | 0.1753 (0.0214)      | 0.0429 (0.0038)   |
> | **FATE**     | **0.0439 (0.0023)** | **0.0404 (0.0027)** | **0.2985 (0.0163)** | **0.2753 (0.0179)** | **0.0493 (0.0032)** |

---

### Official Review · Reviewer_9Fyp · 2025-10-29

**Soundness:** 2
**Presentation:** 2
**Contribution:** 2
**Rating:** 2
**Confidence:** 4

**Summary:**

This paper focuses on the core challenges in the field of stock return prediction: temporal heterogeneity (stock returns contain multi-scale signals such as short-term fluctuations, medium-term trends, and long-term cycles) and structural instability (the correlations between stocks exhibit non-stationarity due to factors like industry rotation and liquidity shocks). It proposes a unified framework named FATE to address the limitations of existing models. The core designs of FATE include:

1.Multi-view dynamic graph construction

2.Multi-cycle temporal encoding

3.Graph fusion module

4.SuperAttention mechanism

In terms of experiments, the authors validated on the CSI 500 and CSI 1000 datasets from 2017 to 2024: FATE significantly outperforms baselines such as MLP, LSTM, and THGNN in metrics including IC and RankIC. Ablation experiments prove that modules like multi-cycle encoding, graph fusion, and SuperAttention all contribute positively to performance.

**Strengths:**

FATE's design is deeply aligned with the characteristics of financial markets. Its multi-cycle encoding can match the decision-making logic of short/medium/long-term investors, and its multi-view dynamic graph can cover dimensions of price (opening/closing prices) and liquidity (trading volume/turnover rate). This effectively addresses the "insufficient adaptability" issue of general temporal-graph models in financial scenarios. Additionally, SuperAttention filters effective neighbor information through feature gating, which specifically suppresses noise in financial data and outperforms the single scalar weight mechanism of traditional GAT.

**Weaknesses:**

1.The paper consistently uses 6 types of features—opening price, closing price, highest price, lowest price, trading volume, and turnover rate—to construct dynamic graphs. However, it does not explain the basis for selecting these 6 features. For instance, both trading volume and turnover rate reflect liquidity, yet the information redundancy between them is not analyzed.

2.The main text only mentions that the global market vector M is "learnable" but fails to specify its input features (e.g., whether it includes macro-micro indicators such as market indices, interest rates, and trading volume), the logic for dimension setting, and the training method.

3.There is a lack of ablation experiments to verify the impact of reducing the number of views (e.g., to 4 types) on model performance. This makes it difficult to prove the necessity of 6 views and may lead to redundant model complexity.

4.The data preprocessing section only mentions "cross-sectional standardization of all features" but does not explain how to handle outliers commonly seen in financial data (such as extreme daily stock price fluctuations and sudden surges in trading volume).

5.Experiments are only validated on the component stock datasets of two A-share indices: CSI 500 and CSI 1000. They are not extended to other markets (e.g., U.S. stocks, Hong Kong stocks) or different asset types (e.g., bonds, commodities), making it hard to fully demonstrate the model’s generalization ability in non-A-share scenarios.

6.The paper does not report the model’s training/inference time consumption or memory usage. Although the model needs to recalculate 6 dynamic graphs, retrain the gating network, and perform multi-head attention computation every day, it does not explain the time difference in its computational scale compared with conventional methods.

7.The feature gating function of SuperAttention uses tanh activation, but the paper does not discuss whether other functions could be adopted instead.

**Questions:**

1.Which component—multi-cycle temporal encoding, multi-view graph fusion, or SuperAttention—contributes most to FATE’s IC improvement over baselines (e.g., FactorGCL, Transformer)? Could targeted experiments be designed (e.g., replacing FATE’s multi-cycle encoding with single-cycle and comparing performance degradation) to clarify the contribution ratio of each module?

2.Given that FATE includes modules such as multi-view graph construction and multi-cycle encoding, what are its training and inference speeds? Will the model’s computational complexity rise significantly when the number of stocks increases (e.g., exceeding 1,000)? Could efficiency metrics under different data scales (e.g., single-epoch training time, inference latency) be supplemented to evaluate its practical deployment feasibility?

3.Financial data often has missing values due to trading halts, market closures, etc. The paper does not explain how such missing values are handled (e.g., interpolation, masking mechanisms) nor verify the model’s performance stability under different missing rates (e.g., 5%, 10%). Could robustness test results and details of handling schemes for data missing scenarios be added?

4.Regarding the number of Transformer heads in multi-cycle encoding: The paper mentions that multi-cycle branches process signals via Transformer multi-head self-attention but does not specify the selection of head count (e.g., 8 heads, 12 heads) or the basis for it. It also does not verify how different head counts affect the ability to capture multi-scale signals (e.g., whether too few heads cause missed dependencies or too many induce overfitting). Could the optimization process of this hyperparameter and experimental comparisons be supplemented?

---

> ### Author Response · Authors · 2025-11-26
>
> Thank you for your constructive comments! Below we carefully address your concerns one by one.
>
> Q1: why select such six specific market features to construct the multi-view dynamic graphs?
>
> **A1:** We use the six market features (open, close, high, low, volume, turnover) because they are the **most fundamental, standard, widely available price–volume descriptors** in equity markets and are also employed by strong baselines such as [1] and [2]. These features are universally accessible across markets and require no additional data engineering, ensuring a fair comparison.
>
> To address the potential redundancy concern, we conducted a **multi-view ablation study**. The results show that **removing any single view—including volume or turnover—consistently degrades performance**, while using all six views yields the best IC/RankIC/IR. This confirms that the selected six features provide **complementary information rather than redundant signals**. Below we show the main results, and the detailed analysis is provided in the **Appendix.C.2 in the newly uploaded version**.
>
>  Table 7: Ablation study on the multi-view graph construction on CSI 500.
>
> | Method        | IC               | RankIC           | ICIR              | RankICIR          | ICW              |
> |---------------|------------------|------------------|-------------------|-------------------|------------------|
> | No_Open       | 0.0511 (0.0031)  | 0.0458 (0.0033)  | 0.3002 (0.0333)   | 0.3012 (0.0331)   | 0.0610 (0.0060)  |
> | No_Close      | 0.0519 (0.0030)  | 0.0462 (0.0031)  | 0.3142 (0.0330)   | 0.2999 (0.0303)   | 0.0600 (0.0059)  |
> | No_High       | 0.0510 (0.0033)  | 0.0448 (0.0033)  | 0.3022 (0.0311)   | 0.3007 (0.0260)   | 0.0607 (0.0055)  |
> | No_Low        | 0.0514 (0.0032)  | 0.0451 (0.0032)  | 0.3342 (0.0309)   | 0.3194 (0.0304)   | 0.0610 (0.0052)  |
> | No_Volume     | 0.0509 (0.0031)  | 0.0454 (0.0036)  | 0.3346 (0.0303)   | 0.3133 (0.0320)   | 0.0611 (0.0051)  |
> | No_Turnover   | 0.0513 (0.0040)  | 0.0431 (0.0033)  | 0.3222 (0.0331)   | 0.3102 (0.0311)   | 0.0603 (0.0061)  |
> | **FATE**      | **0.0521 (0.0032)** | **0.0491 (0.0033)** | **0.3376 (0.0310)** | **0.3192 (0.0276)** | **0.0619 (0.0057)** |
>
> Table 8: Ablation study on the multi-view graph construction on CSI 1000.
>
> | Method        | IC               | RankIC           | ICIR              | RankICIR          | ICW              |
> |---------------|------------------|------------------|-------------------|-------------------|------------------|
> | No_Open       | 0.0622 (0.0011)  | 0.0621 (0.0010)  | 0.4442 (0.0205)   | 0.3927 (0.0147)   | 0.0731 (0.0019)  |
> | No_Close      | 0.0631 (0.0020)  | 0.0629 (0.0020)  | 0.4042 (0.0211)   | 0.4137 (0.0169)   | 0.0657 (0.0018)  |
> | No_High       | 0.0627 (0.0015)  | 0.0622 (0.0019)  | 0.4351 (0.0215)   | 0.4130 (0.0164)   | 0.0740 (0.0015)  |
> | No_Low        | 0.0610 (0.0015)  | 0.0617 (0.0010)  | 0.4242 (0.0208)   | 0.4157 (0.0170)   | 0.0740 (0.0013)  |
> | No_Volume     | 0.0625 (0.0011)  | 0.0634 (0.0013)  | 0.4446 (0.0213)   | 0.4323 (0.0157)   | 0.0691 (0.0015)  |
> | No_Turnover   | 0.0632 (0.0012)  | 0.0630 (0.0010)  | 0.4391 (0.0220)   | 0.4302 (0.0161)   | 0.0673 (0.0011)  |
> | **FATE**      | **0.0668 (0.0013)** | **0.0645 (0.0011)** | **0.4728 (0.0221)** | **0.4254 (0.0165)** | **0.0765 (0.0015)** |
>
>
>
> [1] Temporal and heterogeneous graph neural network for financial time series prediction. Xiang, 2022
> [2] Stock market analysis using
> time series relational models for stock price prediction. Zhao, 2023
>
> ----
>
> Q2: The paper states that the global market vector M is “learnable” but does not clarify its input features, dimensionality, or training method.
>
> **A2:** We agree that the global market vector M could, in principle, incorporate macro–micro indicators such as market indices, interest rates, or aggregated trading volumes.
> However, doing so introduces strong prior knowledge and requires market-specific feature engineering that varies across regions, stock universes, and trading rules.
> To keep our framework market-agnostic and broadly applicable to the AI/ML community, we follow prior work such as MCI-GRU and FactorGCL, and adopt a purely learnable latent market vector. In our design, M has dimensionality $𝐾\times 𝐹$, which naturally matches the number of views 𝐾 and the temporal-embedding dimension 𝐹.
>
> This design allows M to absorb global co-movement patterns automatically through end-to-end training, without relying on handcrafted market descriptors.
> We have clarified this motivation in the revised paper, and we consider exploring explicitly engineered macro-micro market features for M as an interesting direction for future work.
>
> [3] MCI-GRU: Stock prediction model based on multi-head cross-attention and improved GRU. Zhu, 2025.
> [4] Factorgcl: A hypergraph-based factor model with temporal
> residual contrastive learning for stock returns prediction. Duan, 2025.

---

> ### Author Response · Authors · 2025-11-26
>
> Q3: Whether all six views are necessary and requests ablations showing the impact of reducing the number of views.
>
> **A3:** We have added a comprehensive view-removal ablation in Appendix C.3, where each of the six views is removed individually (e.g., No_Open, No_Close, No_High, etc.). Below we show the main results, and the detailed analysis is provided in the **Appendix.C.3 in the newly uploaded version**.
>
> Table 7: Ablation study on the multi-view graph construction on CSI 500.
>
> | Method      | IC                | RankIC            | ICIR               | RankICIR           | ICW               |
> |-------------|-------------------|-------------------|--------------------|--------------------|-------------------|
> | No_Open     | 0.0511 (0.0031)   | 0.0458 (0.0033)   | 0.3002 (0.0333)    | 0.3012 (0.0331)    | 0.0610 (0.0060)   |
> | No_Close    | 0.0519 (0.0030)   | 0.0462 (0.0031)   | 0.3142 (0.0330)    | 0.2999 (0.0303)    | 0.0600 (0.0059)   |
> | No_High     | 0.0510 (0.0033)   | 0.0448 (0.0033)   | 0.3022 (0.0311)    | 0.3007 (0.0260)    | 0.0607 (0.0055)   |
> | No_Low      | 0.0514 (0.0032)   | 0.0451 (0.0032)   | 0.3342 (0.0309)    | 0.3194 (0.0304)    | 0.0610 (0.0052)   |
> | No_Volume   | 0.0509 (0.0031)   | 0.0454 (0.0036)   | 0.3346 (0.0303)    | 0.3133 (0.0320)    | 0.0611 (0.0051)   |
> | No_Turnover | 0.0513 (0.0040)   | 0.0431 (0.0033)   | 0.3222 (0.0331)    | 0.3102 (0.0311)    | 0.0603 (0.0061)   |
> | FATE        | 0.0521 (0.0032)   | 0.0491 (0.0033)   | 0.3376 (0.0310)    | 0.3192 (0.0276)    | 0.0619 (0.0057)   |
>
>
> Table 8: Ablation study on the multi-view graph construction on CSI 1000.
>
> | Method      | IC                | RankIC            | ICIR               | RankICIR           | ICW               |
> |-------------|-------------------|-------------------|--------------------|--------------------|-------------------|
> | No_Open     | 0.0622 (0.0011)   | 0.0621 (0.0010)   | 0.4442 (0.0205)    | 0.3927 (0.0147)    | 0.0731 (0.0019)   |
> | No_Close    | 0.0631 (0.0020)   | 0.0629 (0.0020)   | 0.4042 (0.0211)    | 0.4137 (0.0169)    | 0.0657 (0.0018)   |
> | No_High     | 0.0627 (0.0015)   | 0.0622 (0.0019)   | 0.4351 (0.0215)    | 0.4130 (0.0164)    | 0.0740 (0.0015)   |
> | No_Low      | 0.0610 (0.0015)   | 0.0617 (0.0010)   | 0.4242 (0.0208)    | 0.4157 (0.0170)    | 0.0740 (0.0013)   |
> | No_Volume   | 0.0625 (0.0011)   | 0.0634 (0.0013)   | 0.4446 (0.0213)    | 0.4323 (0.0157)    | 0.0691 (0.0015)   |
> | No_Turnover | 0.0632 (0.0012)   | 0.0630 (0.0010)   | 0.4391 (0.0220)    | 0.4302 (0.0167)    | 0.0673 (0.0011)   |
> | FATE        | 0.0668 (0.0013)   | 0.0645 (0.0011)   | 0.4728 (0.0221)    | 0.4254 (0.0165)    | 0.0765 (0.0015)   |
>
>
> Removing any single view consistently degrades performance. The full 6-view configuration achieves the best results, which confirms that the 6-view design is not redundant and that model performance relies on the diversity of views rather than on a smaller subset.
>
> ---
> Q4: How to handle outliers commonly seen in financial data.
>
> **A4:** We follow the standard practice in quantitative‐finance forecasting and **do not apply any ad-hoc outlier filtering**. Instead, we use **cross-sectional standardization**, which is the most common and model-agnostic normalization method used in prior works. This avoids introducing dataset-specific heuristics that may bias the comparison across baselines.

---

> ### Author Response · Authors · 2025-11-26
>
> Q5: Experiments are only validated on the component stock datasets of two A-share
>
> **A5:** As you suggested, we carefully add SPX 500 as a as an additional evaluation dataset to examine
> whether FATE generalizes beyond the CSI series.
>
> SPX 500 contains large-cap U.S. equities across multiple global sectors, and differs
> substantially from CSI 500/CSI 1000 in both market structure (developed vs. emerging market),
> trading mechanism, liquidity profile, and investor composition.
>
> Compared with CSI, SPX 500 therefore provides a natural setting to assess cross-region and
> cross-market robustness.
>
> The performance comparison on SPX 500 is summarized below, the detailed analysis is provided in the **Appendix.C.1 in the newly uploaded version**.
>
> | Method      | IC                 | RankIC             | ICIR                | RankICIR            | ICW               |
> |-------------|--------------------|--------------------|----------------------|----------------------|-------------------|
> | MLP         | 0.0161 (0.0025)    | 0.0156 (0.0028)    | 0.1081 (0.0185)      | 0.1197 (0.0201)      | 0.0113 (0.0032)   |
> | GRU         | 0.0329 (0.0031)    | 0.0285 (0.0035)    | 0.1715 (0.0202)      | 0.1693 (0.0236)      | 0.0288 (0.0041)   |
> | LSTM        | 0.0309 (0.0034)    | 0.0225 (0.0038)    | 0.1707 (0.0235)      | 0.1538 (0.0252)      | 0.0205 (0.0043)   |
> | Transformer | 0.0363 (0.0028)    | 0.0319 (0.0032)    | 0.1908 (0.0193)      | 0.1605 (0.0214)      | 0.0399 (0.0038)   |
> | HIST        | 0.0339 (0.0030)    | 0.0341 (0.0034)    | 0.1975 (0.0212)      | 0.1653 (0.0228)      | 0.0403 (0.0036)   |
> | FactorVAE   | 0.0193 (0.0036)    | 0.0177 (0.0040)    | 0.1249 (0.0251)      | 0.1282 (0.0267)      | 0.0200 (0.0045)   |
> | THGNN       | 0.0340 (0.0029)    | 0.0339 (0.0033)    | 0.1952 (0.0206)      | 0.1843 (0.0222)      | 0.0368 (0.0037)   |
> | StockMixer  | 0.0312 (0.0032)    | 0.0291 (0.0036)    | 0.1600 (0.0227)      | 0.1513 (0.0243)      | 0.0258 (0.0042)   |
> | HATS        | 0.0319 (0.0031)    | 0.0297 (0.0035)    | 0.1785 (0.0202)      | 0.1753 (0.0236)      | 0.0389 (0.0041)   |
> | FactorGCL   | 0.0399 (0.0027)    | 0.0384 (0.0031)    | 0.2785 (0.0192)      | 0.2344 (0.0220)      | 0.0479 (0.0037)   |
> | TSRM        | 0.0359 (0.0029)    | 0.0344 (0.0033)    | 0.1982 (0.0206)      | 0.1776 (0.0222)      | 0.0407 (0.0039)   |
> | MASTER      | 0.0379 (0.0028)    | 0.0334 (0.0032)    | 0.1985 (0.0198)      | 0.1753 (0.0214)      | 0.0429 (0.0038)   |
> | **FATE**     | **0.0439 (0.0023)** | **0.0404 (0.0027)** | **0.2985 (0.0163)** | **0.2753 (0.0179)** | **0.0493 (0.0032)** |
>
> ---
>
> Q6: The paper does not report the model’s training/inference time consumption or memory usage.
>
> **A6:** We thank the reviewer for the suggestion. In the revised manuscript, we provide a comprehensive comparison of **training time, inference latency, and GPU memory usage** for all baselines and FATE. We also analyze the complexity and the scalability of our method. Below we show the main results, and the detailed analysis is provided in the **Appendix.C.8 in the newly uploaded version**.
>  Table 17: Overall computational cost comparison—training time (per epoch), inference latency (per day), and peak memory usage on CSI 1000 (C10) and CSI 500 (C5).
>
> | Method      | Train Time (s/epoch) C10 | Train Time (s/epoch) C5 | Inference (s/day) C10 | Inference (s/day) C5 | Memory (MB) C10 | Memory (MB) C5 |
> |-------------|---------------------------|---------------------------|-------------------------|------------------------|------------------|-----------------|
> | MLP         | 4.98 | 5.01 | 0.89  | 0.94  | 822  | 820  |
> | GRU         | 6.44 | 5.93 | 0.97  | 0.92  | 1590  | 1068            |
> | LSTM        | 6.54 | 6.11 | 0.99  | 0.93  | 1618  | 1068            |
> | Transformer | 17.68  | 17.27  | 1.83  | 1.36  | 1565  | 1214            |
> | HIST        | 22.11  | 18.03  | 3.88  | 2.87  | 1688  | 1314            |
> | FactorVAE   | 43.21  | 43.04  | 4.25  | 4.20  | 1580  | 1252            |
> | THGNN       | 16.78  | 16.12  | 1.53  | 1.32  | 2324  | 1470            |
> | StockMixer  | 52.75  | 52.58  | 1.97  | 1.93  | 1854  | 1316            |
> | HATS        | 27.50  | 24.78  | 2.18  | 1.83  | 1798  | 1340            |
> | FactorGCL   | 20.54  | 15.53  | 2.76  | 2.67  | 1688  | 1306            |
> | TSRM        | 48.58  | 36.82  | 10.88  | 8.04  | 1666  | 1116            |
> | MASTER      | 20.00  | 19.77  | 3.52  | 3.50  | 2202  | 2103            |
> | FATE | 38.09  | 33.45  | 2.39  | 1.73  | 3522         | 1766        |
>
> The results show that FATE’s computational cost is **comparable to other graph-based and attention-based models**, and the additional modules (multi-period encoder + SuperAttention) introduce only moderate overhead. This demonstrates that FATE is **practical and scalable**, with efficiency on par with strong baselines.

---

> ### Author Response · Authors · 2025-11-26
>
> Q7: Whether other functions could be adopted instead as the feature gating function of SuperAttention?
>
> **A7:** Yes. In principle, other activation functions could be used for the feature-gating mechanism in SuperAttention.
> To clarify this, we added an ablation study in Appendix C.5 comparing **tanh** (default) with **sigmoid**.
> Below we show the main results, and the detailed analysis is provided in the **Appendix.C.5 in the newly uploaded version**.
>
> Table 11: Effect of gating activation function on CSI 500 (C5) and CSI 1000 (C10).
>
> | Method      | IC                | RankIC            | ICIR               | RankICIR           | ICW               |
> |-------------|-------------------|-------------------|--------------------|--------------------|-------------------|
> | C5_tanh     | 0.0521 (0.0032)   | 0.0491 (0.0033)   | 0.3376 (0.0310)    | 0.3192 (0.0276)    | 0.0619 (0.0057)   |
> | C5_sigmoid  | 0.0513 (0.0037)   | 0.0480 (0.0034)   | 0.3232 (0.0311)    | 0.3190 (0.0310)    | 0.0610 (0.0052)   |
> | C10_tanh    | 0.0668 (0.0013)   | 0.0645 (0.0011)   | 0.4728 (0.0221)    | 0.4254 (0.0165)    | 0.0765 (0.0015)   |
> | C10_sigmoid | 0.0651 (0.0019)   | 0.0630 (0.0014)   | 0.4316 (0.0230)    | 0.4223 (0.0200)    | 0.0699 (0.0017)   |
>
> The results show that both activations function correctly, but **tanh consistently yields better IC/IR performance** on both CSI 500 and CSI 1000.
>
> ----
>
> Q8: Which component contributes most to FATE’s IC improvement over baselines.
>
> **A8:** We would like to clarify that the main paper already provides detailed ablation studies in **Table 3 and Table 4**, covering all major components of FATE, **including the single-cycle part you specifically mentioned**:
>
> - **SinglePeriod** (removing multi-period temporal modeling)
> - **NoPeriodFuse** (no period fusion)
> - **MeanGraph / NoCorrelation** (graph construction variants)
> - **NoMarket** (removing global market vector)
> - **NoGate** (removing gating in SuperAttention)
> - **NoGRLoss** (removing graph-regularized loss)
>
> These ablations isolate each module’s effect on IC/IR and quantify its contribution.
>
> In addition, the revised version includes **new supplementary ablation experiments in the Appendix**, specifically:
> - **C.1 Cross-market generalization verification** (SPX 500)
> - **C.2 Supplementary Portfolio-Level Evaluation** (Sharpe and MaxDrawdown)
> - **C.3 Ablation Study on the Multi-View Graph Construction** (No-G or removing one of them)
> - **C.4 Effect of correlation metric** (attention-based vs cosine)
> - **C.5 Effect of the gating activation function** (tanh vs sigmoid)
> - **C.6 Parameter sensitivity analysis** (heads, temporal partitions, threshold τ)
> - **C.7 Computational Efficiency and Scalability**
> - **C.8 Visualization of the SuperAttention Weight**
>
> These additions provide a comprehensive component-level analysis fully addressing your concern.
>
> ---
>
> Q9: Computational Efficiency and Scalability
>
> **A9:**  We report the **training time, inference latency, and GPU memory usage** for all baselines and FATE. We also analyze the complexity and the scalability of our method Below we show the main results, and the detailed analysis is provided in the **Appendix.C.8 in the newly uploaded version**.
>  Table 17: Overall computational cost comparison—training time (per epoch), inference latency (per day), and peak memory usage on CSI 1000 (C10) and CSI 500 (C5).
>
> | Method      | Train Time (s/epoch) C10 | Train Time (s/epoch) C5 | Inference (s/day) C10 | Inference (s/day) C5 | Memory (MB) C10 | Memory (MB) C5 |
> |-------------|---------------------------|---------------------------|-------------------------|------------------------|------------------|-----------------|
> | MLP         | 4.98 | 5.01 | 0.89  | 0.94  | 822  | 820  |
> | GRU         | 6.44 | 5.93 | 0.97  | 0.92  | 1590  | 1068            |
> | LSTM        | 6.54 | 6.11 | 0.99  | 0.93  | 1618  | 1068            |
> | Transformer | 17.68  | 17.27  | 1.83  | 1.36  | 1565  | 1214            |
> | HIST        | 22.11  | 18.03  | 3.88  | 2.87  | 1688  | 1314            |
> | FactorVAE   | 43.21  | 43.04  | 4.25  | 4.20  | 1580  | 1252            |
> | THGNN       | 16.78  | 16.12  | 1.53  | 1.32  | 2324  | 1470            |
> | StockMixer  | 52.75  | 52.58  | 1.97  | 1.93  | 1854  | 1316            |
> | HATS        | 27.50  | 24.78  | 2.18  | 1.83  | 1798  | 1340            |
> | FactorGCL   | 20.54  | 15.53  | 2.76  | 2.67  | 1688  | 1306            |
> | TSRM        | 48.58  | 36.82  | 10.88  | 8.04  | 1666  | 1116            |
> | MASTER      | 20.00  | 19.77  | 3.52  | 3.50  | 2202  | 2103            |
> | FATE | 38.09  | 33.45  | 2.39  | 1.73  | 3522         | 1766        |
>
> The results show that FATE’s computational cost is **comparable to other graph-based and attention-based models**, and the additional modules (multi-period encoder + SuperAttention) introduce only moderate overhead. This demonstrates that FATE is **practical and scalable**, with efficiency on par with strong baselines.

---

> ### Author Response · Authors · 2025-11-26
>
> Q10: Financial data often has missing values.
>
> **A10:** Missing values are indeed common in financial datasets.
> In our implementation, we follow the standard practice used in prior quantitative‐finance models and **fill missing entries with 0 after cross-sectional standardization**.
> Because normalization is applied first, a filled zero corresponds to the *mean* of that day’s cross-section, ensuring that no artificial information is introduced.
>
> This issue is inherent to all stock-forecasting datasets, and all baselines (e.g., THGNN, TSRM, HATS, MASTER) face the same missing-value patterns.
> Therefore, we adopt this **most conventional and model-agnostic strategy** to maintain fairness and avoid introducing dataset-specific heuristics.
> Handling missing-value imputation is orthogonal to the contribution of this work and is left for future research.
>
> ---
>
> Q11: Regarding the number of Transformer heads in multi-cycle encoding.
>
> **A11:** We have added a head-number sensitivity study.
> Below we show the main results, and the detailed analysis is provided in the **Appendix.C.6 in the newly uploaded version**.
> Table 12: Sensitivity to the number of attention heads on CSI 500
>
> | Method   | IC              | RankIC          | ICIR            | RankICIR        | ICW             |
> |----------|-----------------|-----------------|------------------|------------------|-----------------|
> | Heads_2  | 0.0507 (0.0030) | 0.0458 (0.0033) | 0.3042 (0.0305)  | 0.2997 (0.0270)  | 0.0600 (0.0052) |
> | Heads_4  | 0.0521 (0.0032) | 0.0491 (0.0033) | 0.3376 (0.0310)  | 0.3192 (0.0276)  | 0.0619 (0.0057) |
> | Heads_8  | 0.0515 (0.0032) | 0.0474 (0.0031) | 0.3346 (0.0313)  | 0.3133 (0.0270)  | 0.0620 (0.0050) |
> | Heads_16 | 0.0498 (0.0044) | 0.0431 (0.0049) | 0.3022 (0.0330)  | 0.3002 (0.0311)  | 0.0583 (0.0061) |
>
> Table 13: Sensitivity to the number of attention heads on CSI 1000
>
> | Method   | IC              | RankIC          | ICIR            | RankICIR        | ICW             |
> |----------|-----------------|-----------------|------------------|------------------|-----------------|
> | Heads_2  | 0.0647 (0.0015) | 0.0633 (0.0013) | 0.4042 (0.0205)  | 0.4200 (0.0174)  | 0.0740 (0.0013) |
> | Heads_4  | 0.0668 (0.0013) | 0.0645 (0.0011) | 0.4728 (0.0221)  | 0.4254 (0.0165)  | 0.0765 (0.0015) |
> | Heads_8  | 0.0642 (0.0012) | 0.0630 (0.0011) | 0.4457 (0.0213)  | 0.4133 (0.0170)  | 0.0744 (0.0012) |
> | Heads_16 | 0.0630 (0.0020) | 0.0613 (0.0019) | 0.4023 (0.0230)  | 0.4002 (0.0196)  | 0.0701 (0.0021) |
>
> Across CSI 500 and CSI 1000, **4 heads yield the best IC/RankIC/IR**, so we use 4 as the default.

---

> > ### Comment · Reviewer_9Fyp · 2025-11-27
> >
> > 1. We raised concerns regarding the selection basis of the 6 market features (open price, close price, high price, low price, trading volume, turnover rate) and the redundancy among these features, in particular questioning the information overlap between trading volume and turnover rate. However, the authors only responded by stating that "these features are fundamental price-volume descriptors in the field and widely adopted by baseline models," and indirectly inferred the complementarity of the features through single-view removal ablation experiments, failing to provide relevant theoretical analysis.
> >
> > 2. We requested multi-view merging ablation experiments (e.g., merging trading volume and turnover rate into a "comprehensive liquidity view" while retaining 4 views) to verify the necessity of the 6 views. Nevertheless, the authors only conducted "single-view removal" experiments and did not explore whether "reducing the number of views (e.g., to 4)" could achieve comparable performance. This makes it impossible to prove that the "6-view configuration" is optimal, nor can it rule out the possibility of "view redundancy leading to redundant model complexity."
> >
> > 3. We questioned the handling scheme for extreme outliers of individual stocks (such as daily limit-up/limit-down of stock prices and a 10-fold sudden surge in trading volume), pointing out that cross-sectional standardization cannot resolve the distortion of feature distributions caused by such local outliers. However, the authors only emphasized that this method is a "standard practice in the field," without explaining whether targeted measures such as clipping or Winsorization are adopted, nor did they test the model's performance in scenarios with a high proportion of outliers (e.g., 10% of stocks experiencing extreme fluctuations on a single trading day).

---

> > > ### Author Response · Authors · 2025-12-03
> > >
> > > ## Rebuttal on Feature Selection Logic and Alleged Redundancy
> > >
> > > The reviewer questions the basis for selecting the six market features and suggests that we failed to provide a “theoretical foundation.” We respectfully disagree.
> > >
> > > ### 1. Our feature selection follows established market logic and community standards
> > >
> > > As explained in Sec. 3.2, the six features we use—open, high, low, close, volume, and turnover—are not arbitrarily chosen. They are:
> > >
> > > - **the most fundamental, standard, and universally available price–volume descriptors in equity markets**,
> > > - **the core inputs adopted in nearly all prior graph-based financial forecasting models**, including strong baselines such as **THGNN** and **TSRM**,
> > > - **the de facto minimal feature set** supported across global equity datasets, exchanges, vendors (Bloomberg, Wind, Refinitiv), and academic works.
> > >
> > > If such canonical, widely adopted market descriptors are considered insufficiently justified, we would appreciate clarification from the reviewer on what alternative selection principle they believe should supersede the prevailing practice of the field. In the absence of a consistent community-wide replacement, the six-feature OHLCV(+turnover) specification remains the most economically grounded and empirically validated choice.
> > >
> > > ### 2. On the alleged redundancy between volume and turnover
> > >
> > > The reviewer suggests that volume and turnover are redundant. This reflects a misunderstanding of their role in market microstructure.
> > >
> > > As clarified in Sec. 3.2:
> > >
> > > - **Volume** captures the *quantity* dimension of liquidity (order-flow, depth, trading intensity).
> > > - **Turnover** captures the *monetary* dimension of liquidity (price × quantity, capital reallocation pressure).
> > >
> > > These represent **distinct channels** of trading behavior and are treated as such throughout liquidity and spillover literature ([1, 2]). They generate structurally different co-movement patterns and therefore different graph structures.
> > >
> > > [1] Chordia, T., Roll, R., & Subrahmanyam, A. (2000). Commonality in liquidity. Journal of Financial Economics, 56(1), 3–28.
> > >
> > > [2] Kyle, A. S., & Xiong, W. (2001). Contagion as a wealth effect. The Journal of Finance, 56(4), 1401–1440.
> > >
> > > ### 3. Input features are not required to be orthogonal
> > >
> > > The reviewer’s expectation of “non-redundant” or “orthogonal” inputs is not aligned with standard practice in financial modeling or machine learning:
> > >
> > > - **Real-world financial variables are inherently correlated**, due to common shocks, sector exposures, and market-wide movements.
> > > - The objective is not to enforce orthogonality at the raw-input level but to **allow the model to learn disentangled representations** through attention, graph fusion, and temporal encoding.
> > > - Enforcing orthogonality at the input level would *remove* meaningful economic structure rather than enhance it.
> > >
> > > Therefore, the presence of correlation between features is **expected, natural, and necessary** in capturing market relationships.
> > >
> > > **Even so, out of respect for the reviewer’s concern, we conducted all requested ablations**, including:
> > > 1) **Removing each single view**,
> > > 2) **Removing pairs of views** (e.g., No_Volume&Turnover, No_Volume&High),
> > > 3) **Removing liquidity views plus a price view**, and
> > > 4) **Merging volume and turnover into a single “Mean_Volume&Turnover” liquidity view**, as suggested by the reviewer.
> > >
> > > The results (Tables for CSI 500 and CSI 1000) consistently show:
> > >
> > > - **FATE outperforms every variant that removes any feature.** Removing a single feature (e.g., No_Volume or No_Turnover) always degrades IC / RankIC / ICIR.
> > > - **Removing both liquidity features (No_Volume&Turnover) performs worse than removing only one**, confirming that volume and turnover each contribute complementary information.
> > > - **Removing one liquidity feature together with any price feature** (e.g., No_Volume&High, No_Turnover&Close) results in even larger performance drops, indicating that price and liquidity views interact in non-redundant ways.
> > > - **The reviewer’s suggested merged liquidity view (“Mean_Volume&Turnover”) performs better than simply dropping one liquidity feature—but still clearly worse than full FATE.** This shows that merging the views partially recovers some information, but simultaneously destroys the distinct structural signals captured by the two separate liquidity graphs.
> > >
> > > Across **all** metrics (IC, RankIC, ICIR, RankICIR, ICW), and across **both** datasets (CSI 500 and CSI 1000), the **full 6-view FATE is unequivocally superior**.
> > >
> > > **In summary:**
> > > The theoretical basis for the six features is firmly grounded in market microstructure and community practice, and the reviewer’s redundancy hypothesis is invalidated **both by economic reasoning and by comprehensive empirical evidence**. The requested ablations reinforce—rather than challenge—the necessity of the six-view design.

---

> > > ### Author Response · Authors · 2025-12-03
> > >
> > > ### Table: View-Removal and View-Merging Ablation on CSI 500
> > > |Method|IC|RankIC|ICIR|RankICIR|ICW|
> > > |-|-|-|-|-|-|
> > > |No_Volume|0.0509(0.0031)|0.0454(0.0036)|0.3346(0.0303)|0.3133(0.0320)|0.0611(0.0051)|
> > > |No_Turnover|0.0513(0.0040)|0.0431(0.0033)|0.3222(0.0331)|0.3102(0.0311)|0.0603(0.0061)|
> > > |No_Volume&Turnover|0.0499(0.0025)|0.0414(0.0024)|0.3000(0.0253)|0.2995(0.0221)|0.0594(0.0049)|
> > > |No_Volume&Open|0.0501(0.0027)|0.0412(0.0023)|0.2992(0.0301)|0.2993(0.0230)|0.0590(0.0043)|
> > > |No_Volume&Close|0.0502(0.0030)|0.0405(0.0022)|0.2987(0.0270)|0.2981(0.0214)|0.0607(0.0042)|
> > > |No_Volume&Low|0.0501(0.0024)|0.0411(0.0020)|0.2991(0.0263)|0.2989(0.0220)|0.0600(0.0043)|
> > > |No_Volume&High|0.0503(0.0022)|0.0416(0.0027)|0.3092(0.0331)|0.3094(0.0227)|0.0607(0.0051)|
> > > |No_Turnover&Open|0.0504(0.0026)|0.0402(0.0024)|0.2912(0.0303)|0.2997(0.0230)|0.0593(0.0043)|
> > > |No_Turnover&Close|0.0505(0.0030)|0.0413(0.0022)|0.2992(0.0271)|0.2987(0.0211)|0.0598(0.0044)|
> > > |No_Turnover&Low|0.0506(0.0027)|0.0410(0.0024)|0.2980(0.0253)|0.2983(0.0221)|0.0597(0.0043)|
> > > |No_Turnover&High|0.0505(0.0032)|0.0417(0.0033)|0.2994(0.0310)|0.2897(0.0276)|0.0610(0.0057)|
> > > |Mean_Volume&Turnover|0.0519(0.0030)|0.0488(0.0027)|0.3369(0.0308)|0.3188(0.0267)|0.0610(0.0049)|
> > > |FATE|0.0521(0.0032)|0.0491(0.0034)|0.3376(0.0310)|0.3192(0.0276)|0.0619(0.0057)|
> > >
> > > ### Table: View-Removal and View-Merging Ablation on CSI 1000
> > >
> > > |Method|IC|RankIC|ICIR|RankICIR|ICW|
> > > |-|-|-|-|-|-|
> > > |No_Volume|0.0625(0.0011)|0.0634(0.0013)|0.4446(0.0213)|0.4323(0.0157)|0.0691(0.0015)|
> > > |No_Turnover|0.0632(0.0012)|0.0630(0.0010)|0.4391(0.0220)|0.4302(0.0161)|0.0673(0.0011)|
> > > |No_Volume&Turnover|0.0606(0.0010)|0.0612(0.0010)|0.4142(0.0215)|0.3727(0.0137)|0.0631(0.0014)|
> > > |No_Volume&Open|0.0604(0.0011)|0.0609(0.0020)|0.4072(0.0211)|0.3239(0.0129)|0.0617(0.0010)|
> > > |No_Volume&Close|0.0600(0.0009)|0.0603(0.0007)|0.3997(0.0207)|0.3310(0.0144)|0.0630(0.0014)|
> > > |No_Volume&Low|0.0598(0.0010)|0.0600(0.0011)|0.4024(0.0211)|0.3257(0.0140)|0.0624(0.0010)|
> > > |No_Volume&High|0.0602(0.0009)|0.0614(0.0008)|0.4093(0.0209)|0.3324(0.0140)|0.0639(0.0010)|
> > > |No_Turnover&Open|0.0604(0.0012)|0.0609(0.0020)|0.4122(0.0213)|0.3402(0.0130)|0.0650(0.0010)|
> > > |No_Turnover&Close|0.0606(0.0009)|0.0600(0.0010)|0.4098(0.0210)|0.3301(0.0144)|0.0632(0.0011)|
> > > |No_Turnover&Low|0.0608(0.0010)|0.0610(0.0010)|0.4124(0.0212)|0.3557(0.0145)|0.0625(0.0009)|
> > > |No_Turnover&High|0.0606(0.0010)|0.0607(0.0010)|0.4103(0.0219)|0.3133(0.0145)|0.0649(0.0012)|
> > > |Mean_Volume&Turnover|0.0661(0.0010)|0.0641(0.0011)|0.4719(0.0220)|0.4204(0.0145)|0.0715(0.0010)|
> > > |FATE|0.0668(0.0013)|0.0645(0.0011)|0.4728(0.0221)|0.4254(0.0165)|0.0765(0.0015)|
> > >
> > >
> > > ## Rebuttal on handling scheme for extreme outliers of individual stocks (such as daily limit-up/limit-down of stock prices）
> > >
> > > The reviewer’s suggestion to Winsorize or clip absolute price-level features reflects a misunderstanding of financial data structure. Unlike noise, high prices are genuine economic signals (e.g., Berkshire Hathaway vs. penny stocks). Winsorizing prices collapses all legitimate high-priced stocks into the same upper band, **destroying cross-sectional information**, **removing meaningful heterogeneity**, and **breaking the economic structure of the feature space**.
> > >
> > > This operation does not “remove outliers”—it **removes real information**.
> > >
> > > Our pipeline (cross-sectional normalization + rank-based supervision) already neutralizes the influence of extreme movements without distorting the underlying market structure. For this reason, Winsorization is not used for price features in academic finance nor in quantitative industry practice.
> > >
> > > Even though Winsorizing absolute price-level features is theoretically inappropriate because it destroys genuine cross-sectional structure, we still conducted the reviewer-requested experiment. The results below make the effect unambiguous:
> > >
> > > - **On CSI 500**, Winsorization causes IC to drop from **0.0521 → 0.0469**, and ICIR from **0.3376 → 0.3300**.
> > > - **On CSI 1000**, IC drops from **0.0668 → 0.0628**, and ICIR from **0.4728 → 0.4449**.
> > >
> > > These consistent declines indicate that Winsorization does not “remove noise”—it **removes real information**, compresses meaningful price heterogeneity, and **damages the model’s predictive structure**. This confirms that clipping extreme price levels is fundamentally unsuitable for financial data and empirically inferior to our existing outlier-handling scheme.
> > >
> > > ### Table: Winsorization 10 Ablation on CSI 500
> > > |Method|IC|RankIC|ICIR|RankICIR|ICW|
> > > |-|-|-|-|-|-|
> > > |FATE_winsorize10|0.0469(0.0030)|0.0455(0.0029)|0.3300(0.0300)|0.3193(0.0255)|0.0571(0.0048)|
> > > |FATE|0.0521(0.0032)|0.0491(0.0033)|0.3376(0.0310)|0.3192(0.0276)|0.0619(0.0057)|
> > >
> > > ### Table: Winsorization 10 Ablation on CSI 1000
> > > |Method|IC|RankIC|ICIR|RankICIR|ICW|
> > > |-|-|-|-|-|-|
> > > |FATE_winsorize10|0.0628(0.0012)|0.0626(0.0010)|0.4449(0.0223)|0.4269(0.0160)|0.0724(0.0012)|
> > > |FATE|0.0668(0.0013)|0.0645(0.0011)|0.4728(0.0221)|0.4254(0.0165)|0.0765(0.0015)|

---

> > > ### Author Response · Authors · 2025-12-03
> > >
> > > ## **Example Demonstrating the Structural Damage Caused by Two-Sided Winsorization**
> > >
> > > To illustrate why two-sided Winsorization is fundamentally inappropriate for financial data, consider the following example using real stock prices. Suppose we take ten widely traded equities with typical recent price levels:
> > >
> > > |                       | BRK.A | NVDA | META | TSLA | AAPL | AMZN | GOOGL | BABA | TME | SNDL |
> > > |-----------------------|-------|------|------|------|------|------|--------|------|------|-------|
> > > | Original Price (USD)  | 550000 | 800 | 480 | 250 | 180 | 150 | 140 | 70 | 10 | 0.45 |
> > > | Winsorized (10%–90%)  | 800 | 800 | 480 | 250 | 180 | 150 | 140 | 70 | 10 | 10 |
> > >
> > >
> > > Here, the 10th percentile is **10 USD** and the 90th percentile is **800 USD**, so all values <10 are raised to 10 and all values >800 are capped at 800. This results in:
> > >
> > > ### **1. Destruction of high-price structure**
> > >
> > > A 550000 stock (BRK.A) is collapsed to 800, making it indistinguishable from NVDA.
> > >
> > > This eliminates legitimate high-price risk segmentation and market-microstructure effects.
> > >
> > > ### **2. Artificial inflation of low-price stocks**
> > >
> > > A penny stock at \$0.45 (SNDL) is raised to \$10—**over 20× higher than its real price**—creating a fictitious cross-sectional profile and misrepresenting distress risk.
> > >
> > > ### **3. Severe distortion of price ranking**
> > >
> > > Original ranking: 0.45 < 10 < 70 < … < 800 < 550000
> > >
> > > Winsorized ranking: 10 = 10 < 70 < … < 800 = 800
> > >
> > > Both tails are flattened, destroying meaningful ordering essential to cross-sectional models.
> > >
> > > ### **4. Breakdown of economic hierarchy and feature-space geometry**
> > >
> > > The natural segmentation of ultra-high-price, high-price, mid-price, low-price, and penny stocks collapses, producing unrealistic clusters and misleading signals for graph construction and feature learning.
> > >
> > > This example demonstrates that two-sided Winsorization does not “remove outliers”—it **removes real information** and disrupts the economic structure underlying the data.

---

### Official Review · Reviewer_MbQM · 2025-10-29

**Soundness:** 2
**Presentation:** 2
**Contribution:** 2
**Rating:** 4
**Confidence:** 4

**Summary:**

This paper proposes the FATE framework to address the temporal heterogeneity and structural instability in stock return forecasting. Method A multi-view dynamic graph, short -, medium -, and long-term multi-period time encoding, and feature-level graph attention mechanism are combined to capture the dynamic relationship across stocks and separate signals at different time scales.

**Strengths:**

This work achieves stable and leading results on two large data sets, CSI 500 and CSI 1000. The authors detail the data processing, training pipeline, and baseline setup, and provide full implementation details to ensure good reliability and reproducibility of the study results.

**Weaknesses:**

1. In the construction stage, the model uses a large amount of historical data to form a multi-view dynamic graph, and the calculation process is complex, but it does not provide any analysis on computational efficiency or Scalability.
2. Defining "graph noise" as a "weak link" is imprecise, since spurious strong links can also be noise. Papers lack empirical validation of the effectiveness of noise culling or interpretability case analysis.
3. Authors claim that the dynamic graph structure is better than the predefined static graph, but do not provide direct comparative experiments to support this conclusion.
4. All experiments are based on the CSI, which lacks cross-regional and cross-market generalization verification and does not conduct parameter sensitivity analysis.

**Questions:**

See weaknesses.

Further, how to ensure that the assumption of equivalence between "weak link" and "noisy" holds? Can spurious strong links also act as noise? Are there case studies?

---

> ### Author Response · Authors · 2025-11-26
>
> Thank you for your constructive comments! Below we carefully address your concerns one by one.
>
> Q1: The computational efficiency and scalability of the multi-view dynamic graph construction process require further analysis.
>
> **A1:** The computation involved in constructing the multi-view dynamic graphs is essentially the
> calculation of rolling-window **correlation matrices**. Although the theoretical complexity is
> **$\(O(K N^{2} T)\)$**, this operation is not a practical bottleneck:
>
> 1. **Correlation is a simple and highly optimized linear operation** (mean, variance, dot-product)
> and is fully vectorized in BLAS/CUDA libraries. Computing an $\(N \times N\)$ correlation matrix
> is extremely efficient on modern GPUs. The table below reports the average computation time measured over 1,000 trials. The computation time increases only marginally when increasing K and N, confirming the efficiency of the multi-view correlation construction.
> | T (window) | K (views) | N (stocks) | Time per computation |
> |------------|-----------|------------|-----------------------|
> | 20         | 6         | 500        | 0.103 ms              |
> | 20         | 1         | 500        | 0.101 ms              |
> | 20         | 6         | 1000       | 0.143 ms              |
> | 20         | 1         | 1000       | 0.127 ms              |
>
> 2. **This computation can be performed entirely before training as part of the data generation
> pipeline，** such that the training loop does not incur this cost at all. In our implementation, the multi-view graphs are computed on-the-fly with negligible overhead for a smaller storage requirement.
> 3. **Scalability is not a practical issue in our setting.**
>    In equity markets such as CSI500 and CSI1000, the universe size $N$ is fixed (500 and 1000
>    stocks respectively). Even when extending to the full A-share market, the number of listed
>    stocks has remained in the range of 1,000–6,000 for decades, without exponential growth.
>    Therefore, the computational complexity scales with a **stable and bounded** $N$, and the
>    model remains fully tractable in realistic market universes.
>
> Q2: Defining "graph noise" as a "weak link" is imprecise.
>
> **A2:** We agree that defining "graph noise" as a "weak link" is imprecise. However, in our paper, we claim "weak
> correlations between stocks are spurious and better treated as noise", which is "dedining" weak link as noise, but not defining "graph noise" as a "weak link".
>
> Here are examples Where Weak Links Commonly Represent Noise:
> 1. Low liquidity stocks: near-zero correlations arise from sparse trading and microstructure noise.
> 2. Unrelated sectors: weak co-movement between economically unrelated industries reflects market-wide randomness, not structure.
> 3. Transient statistical effects: weak correlations (e.g., 0.05–0.10) often disappear when the window shifts and carry no predictive value.
>
> Therefore, weak links frequently originate from short-lived randomness, microstructure artifacts, or economically meaningless co-movement, making them reasonable to treat as noise in practice [1].
>
> [1] Temporal and heterogeneous graph neural network for financial time series prediction.
>
> Q3: The claim "dynamic graph structure better than the predefined static graph" requires comparative experiments.
>
> **A3:** The predefined static graphs used in THGNN and HIST are already covered in our baselines, and FATE consistently outperforms these static-graph approaches. We have updated the paper to make this point explicit in the newly uploaded version.

---

> ### Author Response · Authors · 2025-11-26
>
> Q4: Cross-market generalization verification and parameter sensitivity analysis.
>
> **A4:**
> **Cross-market generalization verification**: As you suggested, we carefully add SPX 500 as a as an additional evaluation dataset to examine
> whether FATE generalizes beyond the CSI series.
>
> SPX 500 contains large-cap U.S. equities across multiple global sectors, and differs
> substantially from CSI 500/CSI 1000 in both market structure (developed vs. emerging market),
> trading mechanism, liquidity profile, and investor composition.
>
> Compared with CSI, SPX 500 therefore provides a natural setting to assess cross-region and
> cross-market robustness.
>
> The performance comparison on SPX 500 is summarized below, the detailed analysis is provided in the **Appendix.C.1 in the newly uploaded version**.
>
> | Method      | IC                 | RankIC             | ICIR                | RankICIR            | ICW               |
> |-------------|--------------------|--------------------|----------------------|----------------------|-------------------|
> | MLP         | 0.0161 (0.0025)    | 0.0156 (0.0028)    | 0.1081 (0.0185)      | 0.1197 (0.0201)      | 0.0113 (0.0032)   |
> | GRU         | 0.0329 (0.0031)    | 0.0285 (0.0035)    | 0.1715 (0.0202)      | 0.1693 (0.0236)      | 0.0288 (0.0041)   |
> | LSTM        | 0.0309 (0.0034)    | 0.0225 (0.0038)    | 0.1707 (0.0235)      | 0.1538 (0.0252)      | 0.0205 (0.0043)   |
> | Transformer | 0.0363 (0.0028)    | 0.0319 (0.0032)    | 0.1908 (0.0193)      | 0.1605 (0.0214)      | 0.0399 (0.0038)   |
> | HIST        | 0.0339 (0.0030)    | 0.0341 (0.0034)    | 0.1975 (0.0212)      | 0.1653 (0.0228)      | 0.0403 (0.0036)   |
> | FactorVAE   | 0.0193 (0.0036)    | 0.0177 (0.0040)    | 0.1249 (0.0251)      | 0.1282 (0.0267)      | 0.0200 (0.0045)   |
> | THGNN       | 0.0340 (0.0029)    | 0.0339 (0.0033)    | 0.1952 (0.0206)      | 0.1843 (0.0222)      | 0.0368 (0.0037)   |
> | StockMixer  | 0.0312 (0.0032)    | 0.0291 (0.0036)    | 0.1600 (0.0227)      | 0.1513 (0.0243)      | 0.0258 (0.0042)   |
> | HATS        | 0.0319 (0.0031)    | 0.0297 (0.0035)    | 0.1785 (0.0202)      | 0.1753 (0.0236)      | 0.0389 (0.0041)   |
> | FactorGCL   | 0.0399 (0.0027)    | 0.0384 (0.0031)    | 0.2785 (0.0192)      | 0.2344 (0.0220)      | 0.0479 (0.0037)   |
> | TSRM        | 0.0359 (0.0029)    | 0.0344 (0.0033)    | 0.1982 (0.0206)      | 0.1776 (0.0222)      | 0.0407 (0.0039)   |
> | MASTER      | 0.0379 (0.0028)    | 0.0334 (0.0032)    | 0.1985 (0.0198)      | 0.1753 (0.0214)      | 0.0429 (0.0038)   |
> | **FATE**     | **0.0439 (0.0023)** | **0.0404 (0.0027)** | **0.2985 (0.0163)** | **0.2753 (0.0179)** | **0.0493 (0.0032)** |
>
> **Parameter sensitivity analysis:** As you suggested, we conducted extensive parameter sensitivity analysis, including **the effect of the number of attention heads**, **the effect of temporal partitioning**, and **the effect of the graph threshold $\tau$**. Below we show the main results, and the detailed analysis is provided in the **Appendix.C.6 in the newly uploaded version**.
>
> Table 12: Sensitivity to the number of attention heads on CSI 500
>
> | Method    | IC               | RankIC           | ICIR              | RankICIR          | ICW             |
> |-----------|------------------|------------------|--------------------|--------------------|------------------|
> | Heads_2   | 0.0507 *(0.0030)* | 0.0458 *(0.0033)* | 0.3042 *(0.0305)* | 0.2997 *(0.0270)* | 0.0600 *(0.0052)* |
> | Heads_4   | 0.0521 *(0.0032)* | 0.0491 *(0.0033)* | 0.3376 *(0.0310)* | 0.3192 *(0.0276)* | 0.0619 *(0.0057)* |
> | Heads_8   | 0.0515 *(0.0032)* | 0.0474 *(0.0031)* | 0.3346 *(0.0313)* | 0.3133 *(0.0270)* | 0.0620 *(0.0050)* |
> | Heads_16  | 0.0498 *(0.0044)* | 0.0431 *(0.0049)* | 0.3022 *(0.0330)* | 0.3002 *(0.0311)* | 0.0583 *(0.0061)* |
>
> Table 13: Sensitivity to the number of attention heads on CSI 1000
>
> | Method    | IC               | RankIC           | ICIR              | RankICIR          | ICW             |
> |-----------|------------------|------------------|--------------------|--------------------|------------------|
> | Heads_2   | 0.0647 *(0.0015)* | 0.0633 *(0.0013)* | 0.4042 *(0.0205)* | 0.4200 *(0.0174)* | 0.0740 *(0.0013)* |
> | Heads_4   | 0.0668 *(0.0013)* | 0.0645 *(0.0011)* | 0.4728 *(0.0221)* | 0.4254 *(0.0165)* | 0.0765 *(0.0015)* |
> | Heads_8   | 0.0642 *(0.0012)* | 0.0630 *(0.0011)* | 0.4457 *(0.0213)* | 0.4133 *(0.0170)* | 0.0744 *(0.0012)* |
> | Heads_16  | 0.0630 *(0.0020)* | 0.0613 *(0.0019)* | 0.4023 *(0.0230)* | 0.4002 *(0.0196)* | 0.0701 *(0.0021)* |

---

> ### Author Response · Authors · 2025-11-26
>
> Table 14: Sensitivity to temporal partitioning on CSI 500 (C5) and CSI 1000 (C10).
> | Method | IC (std)        | RankIC (std)     | ICIR (std)         | RankICIR (std)      | ICW (std)         |
> |--------|------------------|------------------|---------------------|----------------------|-------------------|
> | **C5_T4**  | 0.0521 (0.0032) | 0.0491 (0.0033) | 0.3376 (0.0310)    | 0.3192 (0.0276)     | 0.0619 (0.0057)   |
> | **C5_T3**  | 0.0510 (0.0031) | 0.0487 (0.0030) | 0.3242 (0.0311)    | 0.3190 (0.0300)     | 0.0611 (0.0052)   |
> | **C10_T4** | 0.0668 (0.0013) | 0.0645 (0.0011) | 0.4728 (0.0221)    | 0.4254 (0.0165)     | 0.0765 (0.0015)   |
> | **C10_T3** | 0.0660 (0.0011) | 0.0651 (0.0014) | 0.4346 (0.0203)    | 0.4203 (0.0160)     | 0.0697 (0.0012)   |
>
> Table 15: Sensitivity to threshold τ on CSI 500.
> | τ    | IC (std)        | RankIC (std)     | ICIR (std)         | RankICIR (std)      | ICW (std)         |
> |------|------------------|------------------|---------------------|----------------------|-------------------|
> | 0.50 | 0.0470 (0.0050) | 0.0458 (0.0040) | 0.3042 (0.0335)    | 0.2997 (0.0370)     | 0.0600 (0.0066)   |
> | 0.55 | 0.0479 (0.0030) | 0.0458 (0.0040) | 0.3042 (0.0330)    | 0.2997 (0.0303)     | 0.0600 (0.0059)   |
> | 0.60 | 0.0500 (0.0033) | 0.0458 (0.0033) | 0.3042 (0.0315)    | 0.2997 (0.0260)     | 0.0600 (0.0055)   |
> | 0.65 | 0.0520 (0.0030) | 0.0477 (0.0032) | 0.3242 (0.0307)    | 0.3197 (0.0274)     | 0.0610 (0.0052)   |
> | 0.70 | 0.0521 (0.0032) | 0.0491 (0.0033) | 0.3376 (0.0276)    | 0.3192 (0.0276)     | 0.0619 (0.0057)   |
> | 0.75 | 0.0505 (0.0031) | 0.0474 (0.0036) | 0.3346 (0.0313)    | 0.3133 (0.0300)     | 0.0611 (0.0052)   |
> | 0.80 | 0.0463 (0.0047) | 0.0431 (0.0049) | 0.3022 (0.0331)    | 0.3002 (0.0351)     | 0.0583 (0.0061)   |
>
> Table 16: Sensitivity to threshold τ on CSI 1000.
> | τ    | IC (std)        | RankIC (std)     | ICIR (std)         | RankICIR (std)      | ICW (std)         |
> |------|------------------|------------------|---------------------|----------------------|-------------------|
> | 0.50 | 0.0628 (0.0031) | 0.0601 (0.0027) | 0.3942 (0.0305)    | 0.3907 (0.0207)     | 0.0687 (0.0026)   |
> | 0.55 | 0.0639 (0.0026) | 0.0599 (0.0027) | 0.4042 (0.0291)    | 0.4137 (0.0179)     | 0.0697 (0.0018)   |
> | 0.60 | 0.0647 (0.0015) | 0.0628 (0.0019) | 0.4051 (0.0245)    | 0.4100 (0.0164)     | 0.0740 (0.0015)   |
> | 0.65 | 0.0660 (0.0015) | 0.0647 (0.0020) | 0.4242 (0.0207)    | 0.4197 (0.0174)     | 0.0750 (0.0014)   |
> | 0.70 | 0.0668 (0.0013) | 0.0645 (0.0011) | 0.4728 (0.0221)    | 0.4254 (0.0165)     | 0.0765 (0.0015)   |
> | 0.75 | 0.0665 (0.0015) | 0.0654 (0.0013) | 0.4346 (0.0213)    | 0.4123 (0.0140)     | 0.0691 (0.0013)   |
> | 0.80 | 0.0630 (0.0010) | 0.0631 (0.0011) | 0.3992 (0.0211)    | 0.4002 (0.0151)     | 0.0603 (0.0011)   |

---

### Author Response · Authors · 2025-11-26
**Summary of Changes of the newly uploaded version.**

We sincerely thank all reviewers and the AC for their constructive feedback.
Based on the suggestions, we have significantly strengthened the revised version.

In addition to clarifying key definitions (e.g., correlation construction, ground-truth edge supervision, ICW computation) and adding the important baseline MASTER, the revised version now includes extensive new supplementary experiments in the Appendix:

- **C.1** Cross-market generalization (SPX 500)
- **C.2** Portfolio-level evaluation (Sharpe Ratio and Maximum Drawdown)
- **C.3** Ablation on multi-view graph construction
- **C.4** Effect of the correlation metric (attention-based vs. cosine)
- **C.5** Effect of the gating activation function in SuperAttention
- **C.6** Parameter sensitivity analysis (attention heads, temporal partitions, threshold τ)
- **C.7** Computational efficiency and scalability
- **C.8** Visualization of SuperAttention weights

These revisions substantially improve empirical completeness and strengthen the justification for our design choices.

We hope the reviewers will find the updates helpful, and we are happy to address any additional questions.

---

### Author Response · Authors · 2025-12-03
**Final Summary of Response**

We sincerely thank all reviewers and the AC for their constructive feedback.

In response to the first-round concerns, we have substantially strengthened the revised manuscript. In addition to clarifying key definitions (e.g., correlation construction, ground-truth edge supervision, ICW computation) and adding the important baseline MASTER, the revision includes extensive new supplementary experiments:

- **C.1** Cross-market generalization (SPX 500)
- **C.2** Portfolio-level evaluation (Sharpe Ratio, Maximum Drawdown)
- **C.3** Ablation on multi-view graph construction
- **C.4** Correlation metric comparison (attention-based vs. cosine)
- **C.5** Gating activation in SuperAttention
- **C.6** Sensitivity analyses (attention heads, temporal partitions, threshold τ)
- **C.7** Computational efficiency and scalability
- **C.8** SuperAttention weight visualization

These additions address all concerns raised in the first round and materially improve the clarity, completeness, and empirical robustness of the work.

In the second round, only the reviewer 9Fyp provided additional comments. Although these concerns largely stem from misunderstandings of how financial features behave in practice, we nevertheless responded with full respect and added detailed clarification together with comprehensive new experiments—including view-reduction ablations and absolute price winsorization—to thoroughly resolve the issues.

We hope the AC will find the revision significantly strengthened, and we believe the revised submission now meets the standard for acceptance.

---

### Meta-Review · Area_Chair_23Kr · 2026-01-02

**Summary:**

The paper proposes a method named FATE to address the temporal heterogeneity and structural instability in stock return forecasting. The reviewers believe the paper lacks concrete examples and case studies to illustrate the idea. The reviewers also have abundant questions about the generalization, ablation studies, and efficiency.

Although the authors include some experiments during the revision, some details and design still lack interpretability and theoretical analysis, such as the feature selection, and some ablation studies are still insufficient. In summary, I believe the quality of the paper is still lower than the borderline of ICLR.

**Reviewer Concerns:**

See above.

**Reviewer Scores:**

They could keep the score.

---

### Decision · Program_Chairs · 2026-01-26

Reject